# Cataract Surgery by Intraoperative Surface Irrigation with 0.25% Povidone–Iodine

**DOI:** 10.3390/jcm10163611

**Published:** 2021-08-16

**Authors:** Hiroyuki Shimada, Hiroyuki Nakashizuka

**Affiliations:** Department of Ophthalmology, School of Medicine, Nihon University, 1-6 Surugadai, Kanda, Chiyodaku, Tokyo 101-8309, Japan; nkshizuk@gmail.com

**Keywords:** normal flora, border control, cataract surgery, endophthalmitis, ocular surface irrigation, ocular toxicity, povidone-iodine, drug-resistant bacteria, safe concentration

## Abstract

Postoperative endophthalmitis after cataract surgery is typically caused by the patient’s own conjunctival normal bacterial flora. A three-step approach is recommended to prevent endophthalmitis: (1) “border control” to prevent microorganisms from entering the eye by disinfecting the ocular surface is the most important measure; (2) bacteria that have gained access into the anterior chamber are reduced by irrigation; (3) bacteria remaining in the anterior chamber and vitreous at the end of surgery are controlled by antibacterial drugs. We have devised a method, “the Shimada technique”, for irrigating the ocular surface with povidone-iodine, a disinfectant with potent microbicidal effect and established effective and safe concentrations for eye tissues. Povidone-iodine exhibits a bactericidal effect for a wide concentration range of 0.005–10%, but 0.1% povidone-iodine has the highest activity and requires the shortest time of only 15 s to achieve microbicidal effect. When used to irrigate the ocular surface every 20–30 s during cataract surgery, 0.25% povidone-iodine is conceivably diluted to around 0.1%. Irrigation with 0.25% povidone-iodine during cataract surgery significantly reduced bacteria contamination rate in the anterior chamber compared with saline (*p* = 0.0017) without causing corneal endothelial damage.

## 1. Introduction

Cataract surgery is the most common surgery performed in ophthalmology. With a global cataract prevalence of approximately 50% among adults over the age of 50, cataract surgery is the most common surgical procedure performed in developed countries. Of the overall 32.4 million blind and 191 million visually impaired reported in 2010, 10.8 million people were blind and 35.1 million were visually impaired due to cataracts [1]. Among the complications of cataract surgery, postoperative endophthalmitis is one of the most serious conditions [2], and prevention of postoperative endophthalmitis is an important issue. Postoperative endophthalmitis after cataract surgery is typically caused by the patient’s own normal conjunctival bacterial flora [3]. The current problem is an increase of multidrug-resistant bacteria and fungi causing endophthalmitis [4,5]. This article introduces the concept of a “three-step approach” to prevent postoperative endophthalmitis. Among the “three-step approach”, the “border control” that prevents the normal bacterial flora on the ocular surface from entering the eye is the most important. As a “border control” method, we explain how to clean the eye surface every 20–30 s during cataract surgery with 0.25% povidone-iodine instead of washing with physiological saline.

## 2. Border Control

In a pandemic, once the causative bacteria or viruses successfully gain a foothold in a country, it is difficult to control and eliminate them [6,7,8,9], which makes us keenly aware of the importance of “border control”. Closing the borders, strengthening quarantine measures at airports and ports and blocking the flow of people infected with the bacteria or viruses from entering one’s country are very useful measures. Regarding the threat and control of infection, both countries surrounded by sea and countries bordering other countries are in a situation not unlike that of an eyeball surrounded by surface fluid.

As humans pass through the birth canal at birth, they become contaminated by bacteria that promptly colonize the skin, mucous membranes and surface of the gastrointestinal tract. Many of the normal bacterial flora are non-pathogenic [10]. Since the conjunctiva is constantly washed by tears that are rich in lysozyme [11], the number of bacteria inhabiting the conjunctiva is far less than that on the skin and gastrointestinal tract. The normal bacterial flora on the conjunctiva forms a “natural barrier” against the invasion of pathogens. The species and frequencies of the bacteria in the flora vary depending on geographical location and are affected by the use of antibiotics [12,13,14].

When the intestinal wall is damaged, the bacterial flora in the gastrointestinal tract spread to the abdominal cavity, causing peritonitis [15]. Even non-pathogenic bacteria in the conjunctival flora become pathogenic when they are transferred into the eye. The transformation of non-pathogenic normal bacterial flora of the conjunctiva to pathogenic bacteria causing infection has been explained by the differences in the immunological state between the conjunctiva and the intraocular environment. For this reason, intraocular procedures including cataract surgery, vitrectomy and intravitreal injection that are performed while communicating the extraocular with the intraocular environment always carry the risk of endophthalmitis. Therefore, during intraocular surgery, it is important to implement “border control” measures to prevent not only bacteria but also fungi, viruses and other microbes from entering the eye.

## 3. Increase of Multidrug-Resistant Bacteria and Fungi Causing Endophthalmitis

Endophthalmitis is one of the most serious and devastating conditions within the eye globe, which may lead to irreversible blindness in the affected eye. It is a purulent inflammation of the intraocular fluids comprising the vitreous and the aqueous humor. Although the etiology of this condition can be both endogenous and exogenous, it is mostly secondary to intraocular surgeries, injections or penetrating ocular trauma [16]. The incidence of endophthalmitis after cataract surgery is low with reports of 0.076% [17], 0.02% [18] and 0.048% [19]. Even with appropriate treatments, the final visual acuity is poor, and reported to be less than 20/200 (35.7%) or 20/200 to 20/40 (36.7%) [19]. Given the risk of blindness in many patients with endophthalmitis worldwide, prevention of this complication is a pressing issue.

In a report from India, endophthalmitis was caused by multidrug-resistant bacteria, including vancomycin resistance, in 42 of 807 eyes (5.2%) [20]. Since visual prognosis of endophthalmitis caused by multidrug-resistant bacteria tends to be poor due to the lack of effective antibiotics [20], prevention and treatment of endophthalmitis caused by these bacteria must be addressed. There is also an increase in frequency of endophthalmitis caused by fungi that are resistant to antibiotics, including *Candida albicans* [21] and filamentous fungi [22]. Therefore, it is necessary to prevent endophthalmitis using drugs or other agents with a wide antimicrobial spectrum against also multidrug-resistant bacteria and fungi.

## 4. Factors for Emergence of Multidrug Resistant Bacteria

Three major practices that may lead to the emergence of drug-resistant bacteria have been proposed [23].
(1)Long-term use of broad-spectrum antibiotics without identifying the causative organism(2)Long-term use of antibiotics after infectious disease has been cured(3)Preoperative administration of antibiotics, called infection prophylaxis or preoperative disinfection

Many different practice patterns of disinfection have been reported around the world [24,25]. In particular, the use of prophylactic intracameral antibiotics varies widely among countries and facilities. In Japan, instillation of antibiotic eye drops four times a day from 1–3 days before surgery remains the mainstay of preoperative disinfection before cataract surgery [26]. 

Bacteria are detected in the conjunctiva of healthy people at a frequency of 82.1%, although the number is small (Figure 1). Preoperative instillation of antibiotic eye drops reduces the frequency to 74.6% [27]. Despite this reduction, the prophylactic effect against infection is small. Furthermore, this method has the risk of selecting resistant bacteria. Before cataract surgery, cleaning the eyelid and conjunctiva with povidone-iodine transiently kills the conjunctival bacterial flora. However, during draping and placement of the eyelid speculum, bacteria reappear on the conjunctiva. Indeed, a study demonstrates that even with preoperative washing with povidone-iodine, bacteria are detected at a frequency of 5.5% on the ocular surface at the beginning of cataract surgery [28]. If these bacteria enter the anterior chamber during surgery, there is a risk of developing endophthalmitis postoperatively. After cataract surgery, the same antibiotic eye drops used before surgery are often prescribed for 1–4 weeks. Continuous use of the same antibiotic eye drops induces growth of bacterial species resistant to the eye drops by the microbial substitution phenomenon [29,30,31]. When administration of postoperative antibiotic eye drops is finished, instead of recovery of the bacterial flora to the preoperative state, there is an increase of resistant bacteria. 

Therefore, if bacteria that emerge on the ocular surface during ocular surgery are killed constantly, the practice of preoperative antibiotic prophylaxis can be stopped [29] and the risk of proliferation of resistant bacteria can be reduced [30,31] (Figure 2). Postoperative antibiotic eye drops are expected to penetrate the anterior chamber and exhibit antibacterial effects. In many countries and facilities, postoperative antibiotic eye drops are usually used from less than 7 days to more than 30 days (median 7–13 days) [25]. Shortening the period of using postoperative antibiotic eye drops as far as possible is necessary to prevent the increase of resistant bacteria [32].

## 5. Intraocular Migration of Bacteria during Cataract Surgery

A study using molecular epidemiology including identification of bacterial species and genetical analyses has provided evidence that postoperative endophthalmitis develops when bacteria of the normal flora on the eyelid and conjunctiva gain access into the eye during surgery [33]. To visualize the process of bacterial migration during intraocular surgery, a study was conducted using a porcine eye model [34] by applying fluoresbrite carboxylate microspheres (1 µm; 2.5% Solids-Latex) having the same size as *Staphylococcus* on the conjunctiva, sparing the circumference of the eyeball. Cataract surgery was performed with fluoresbrite microspheres spread on the ocular surface. When the intraocular lens was inserted using an injector (Figure 3A), the microspheres adhered to the tip of the injector and, together with the ophthalmic viscosurgical device, were introduced into the anterior chamber in the shape of a balloon (Figure 3B). The microspheres then spread to surround the anterior and posterior surfaces of the intraocular lens (Figure 3C). When the intraocular lens opened, the microspheres distributed in the whole anterior chamber. Since the microspheres and the ophthalmic viscosurgical device were mixed together in the anterior chamber, even when the anterior chamber was washed, the microspheres on the iris and posterior surface of the cornea remained. As the microspheres were also found on the posterior surface of the intraocular lens, it was necessary to wash the posterior surface of the lens (Figure 3D). At the end of surgery, a small amount of fluoresbrite microspheres also flowed into the vitreous. In the case that posterior capsule rupture or anterior hyaloid membrane tear occurs during cataract surgery, a large amount of the fluoresbrite microspheres that entered the anterior chamber is expected to flow into the vitreous body due to increased ocular pressure [34].

## 6. Prevention of Endophthalmitis during Cataract Surgery: Three-Step Approach

From the experimental results of intraocular migration of bacteria during cataract surgery, we propose a three-step approach to prevent endophthalmitis after cataract surgery. 

**First step** (Figure 4A): The most important measure in preventing endophthalmitis is to prevent bacteria present on the ocular surface from entering the eye. Currently, many surgeons irrigate the eye surface with saline during cataract surgery. This procedure reduces the number of bacteria on the eye surface by flushing them from the operative field. Many types of microbes including general bacteria, drug-resistant bacteria, fungi and viruses on the surface of the eye can potentially invade the eye. A useful preventive approach is the so-called “border control” measure, which is to repeatedly irrigate the surface of the eye with an aqueous solution of wide-spectrum antimicrobial agent to achieve an almost microbe-free ocular surface during surgery.

**Second step** (Figure 4B): Even when the “border control” measure is taken, there is still a risk that bacteria on the ocular surface may enter the eye at the time of insertion of the intraocular lens. As a preventive measure, it is important to clean not only the anterior chamber but also the posterior surface of the intraocular lens after inserting the intraocular lens, in order to reduce the bacterial load [17,35]. 

**Third step** (Figure 4C): Even in cataract surgery performed without surgical complications, bacteria in the anterior chamber may flow into the vitreous body due to the increase in anterior chamber pressure during surgery [34]. The occurrence of complications including posterior capsule rupture, tear of zonule of Zinn and vitreous prolapse further increases the risk of migration of bacteria into the vitreous [17,19]. Since the bacteria in the vitreous cannot be removed by washing, there is no other way but to eliminate them by administering antibacterial drugs. The use of antibiotics varies depending on countries, institutions and surgeons, but not using postoperative antibiotics would increase the risk of endophthalmitis [17]. Many surgeons use intravenous or oral antibiotics to reduce the number of bacteria. Intravenous injection of antibiotics has the advantage of achieving higher blood concentrations than oral administration, and consequently more effective intraocular drug delivery. However, as intravenous injection of antibiotics occasionally causes adverse effects such as drug-induced anaphylactic shock, many surgeons choose the oral route of administration. A number of reports have shown that intracameral injection of antibiotics during cataract surgery effectively reduces many of the bacteria that have entered the anterior chamber [36,37,38]. However, since vitreous penetration is low, the effect of antibiotics on the bacteria that have invaded the vitreous cannot be expected [39,40].

The point that deserves most attention is that the microorganisms that potentially gain access into the anterior chamber and vitreous include drug-resistant bacteria and fungi. Therefore, the key approach to prevent postoperative endophthalmitis is perhaps the “border control” measure of repeatedly irrigating the ocular surface with an aqueous solution of agents with wide-spectrum antimicrobial activities during cataract surgery, aiming to achieve an almost microbe-free ocular surface during surgery.

## 7. Povidone-Iodine as a “Border Control” Measure

Disinfectants are classified into three levels according to characteristics including the antimicrobial spectrum: high-level, intermediate-level and low-level (Figure 5) [41]. Glutaraldehyde is a high-level disinfectant that can kill all microorganisms, but its use is limited to the disinfection of instruments [42]. 

Hypochlorous acid, which belongs to the intermediate-level disinfectants, is an ingredient of chlorine bleach and is not suitable for biological use [43]. Ethanol is used to disinfect the skin but is not suitable for disinfection of the cornea [44]. Ozonated solution (4 ppm) is effective for disinfecting the eye surface [45,46], but has several issues including short half-life, need for ozone generator and damaging to corneal endothelial cells after exposure for over 10 s [47].

Povidone-iodine has been extensively studied, and the effective and safe concentrations have been established. This agent has a wide antimicrobial spectrum, does not induce resistant bacteria, is inexpensive and is used in diverse fields around the world [48]. Povidone-iodine can be used on the skin and mucous membranes and shows marked bactericidal effect upon exposure for 15–180 s. Various studies have confirmed its microbicidal effects against multidrug-resistant bacteria [49], *Candida* spp. [50], viruses [51] and *Acanthamoeba* [52], as well as its anti-biofilm activity [53] (Table 1). As prophylaxis against endophthalmitis, preoperative ocular surface irrigation with povidone-iodine has received higher clinical recommendation than other methods including preoperative topical antibiotics and antibiotic-containing irrigating solutions [54].

Chlorhexidine is a low-level disinfectant and is used for disinfection of the eye surface in intravitreal injection and also used in persons who are hypersensitive to povidone-iodine [55]. It has the advantages of being transparent, inducing little corneal irritation and having almost no odor [56]. However, when placed on the operating table, its transparent appearance may be mistaken for physiological saline and attention should be paid not to mistakenly inject the solution into the eye. A 0.5% solution is used for skin disinfection and 0.05–0.1% for mucous membrane disinfection. Safety for corneal endothelial cells and retina has not been studied. The 90% minimum killing concentration (MKC_90_) against *S. aureus* is 0.5% and MKC_50_ is 0.125% [57]. At the concentration range of 0.05–0.1% used for mucous membrane disinfection, the bactericidal effect on *S. aureus* is mild. 

Given that 0.05–0.1% chlorhexidine disinfection requires a long exposure time to kill *S. aureus*, its antimicrobial spectrum is narrower than that of povidone-iodine, and intraocular safety has not been established, this agent cannot be used for irrigating the ocular surface during surgery.

The antibacterial mechanisms of the commonly used antibiotic eye drops involve selective suppression of bacterial cell membrane components, proteins and DNA [58] (Table 1). However, a problem with these drugs is that drug resistance occurs. They are effective against general bacteria, but not against drug-resistant bacteria, fungi, viruses and *Acanthamoeba*. For eye drops containing high concentrations of antibiotics such as gatifloxacin 0.3% and moxifloxacin 0.5%, an exposure time of 15–60 min is required to achieve bactericidal effect [59]. In one study, the bacteria detection rate in the anterior chamber at the end of cataract surgery was 6.8% when vancomycin and gentamicin were added to the irrigation solution compared with 21.1% when no antibiotics were added, with a significant difference (*p* = 0.0001) [60]. When used at low concentrations in irrigation solution, vancomycin (20 μg/mL, 0.002%) and gentamicin (8 μg/mL: 0.0008%) require more than 140 min to reduce the number of bacteria [61]. From the above findings, antibiotics are not suitable for washing or irrigation of the ocular surface because a long exposure time of 15–60 min is required for bacterial killing, even when high-concentration antibiotic eye drops are used.

## 8. Basic Properties of Povidone-Iodine

Povidone-iodine was developed in 1956 [62]. A 10% povidone-iodine solution contains 1% available iodine. Since iodine has a small molecular weight of 254, it easily penetrates the cornea. The pH of a 5% povidone-iodine solution is 5.0, which is acidic. Povidone-iodine is a chemical complex of polyvinylpyrrolidone and iodine. Iodine oxidizes water, generating ions that act directly on the membrane proteins of bacteria and viruses [39]. Since this effect is not selective for bacteria or other microbes, povidone-iodine also acts directly on the membrane proteins of normal cells. Therefore, using an effective and safe concentration is important.

### 8.1. Effective Concentration

The effective concentration range of povidone-iodine has been reported to be 0.005–10% [63]. There is a misconception that 5% and 10% povidone-iodine has more potent bactericidal effect and requires a shorter exposure time to achieve the bactericidal effect. In fact, iodine does not dissociate readily at high povidone-iodine concentrations but dissociates well as the solution is diluted. The free iodine concentration is 3 ppm for a 10% povidone-iodine solution, 8 ppm for 5% solution, 13 ppm for 1% solution, 24 ppm for 0.1% solution (exposure time for killing *S. aureus*: 15 s) and 13 ppm for 0.01% solution [48,64] (Figure 6). Therefore, 0.1% povidone-iodine has the greatest bactericidal effect. Since iodine is inactivated when reacting with bacteria and organic matter, free iodine needs to be replenished (Table 2). When povidone-iodine is used at high concentrations (2.5–10%), free iodine can be replenished easily from the abundant supply available in the surrounding. At low concentrations (0.1–1.0%) of povidone-iodine, however, the amount of iodine available is small and hence the bactericidal effect does not last, and it is necessary to apply repeatedly to maintain the effect. Although 2.5–10% povidone-iodine requires a longer exposure time for microbicidal effect, it is also longer-acting. For this reason, 2.5–10% povidone-iodine is used in one-time application for eyelid and skin disinfection, and one-time instillation during intravitreal injection. Within the 0.1–1.0% concentration range, 0.05–0.5% povidone-iodine is used for irrigating the ocular surface, as will be discussed later. This low concentration range requires a short exposure time, but the effect is short-acting. Therefore, during cataract surgery, it is necessary to replenish fresh povidone-iodine by irrigating the ocular surface repeatedly every 20–30 s.

### 8.2. Safe Concentration for Irrigating Ocular Surface

A large number of basic studies on povidone-iodine have been conducted (Figure 7). The basic research used 5% Betadine (Alcon Laboratories) with additives [64]. Although povidone-iodine is an acidic solution, tissue toxicity is not related to pH but depends on the concentration of povidone-iodine [48].

In a study on the effect of povidone-iodine on corneal epithelial cells in rabbits, 0.5 mL of 0.5%, 1.0%, 2.5% or 5% povidone-iodine (*n* = 5 each) was instilled into the conjunctival sac [65]. After 30 min, all eyes were stained with fluorescein, the cornea and conjunctiva were observed with a slit lamp, and epithelial damage was graded on a scale of 0–4. All eyes treated with saline were assessed as grade 0, and all eyes treated with 0.5% povidone-iodine as grade 1. At 1%, one eye was grade 1 and four eyes were grade 2. At 2.5%, one eye each was grade 2 and grade 4, and three eyes were grade 3. At 5%, all eyes were grade 4. There were no significant differences in the number of corneal endothelial cells and corneal thickness between groups treated with saline, 0.5% and 1%. With 0.5% povidone-iodine, however, a single instillation of 0.5 mL caused corneal epithelial damage, even though the cornea appeared normal. Therefore, if the ocular surface is washed several times, corneal epithelial damage possibly occurs even at concentrations lower than 0.5%. The povidone-iodine concentration used for washing the ocular surface should be set at 0.5% or below.

The above study also investigated the effect on corneal endothelial cells by injecting 0.05 mL of 0.5%, 1.0%, 1.5% or 2.0% povidone-iodine into the anterior chamber of rabbit eyes (*n* = 8 each) [65]. After 3 days, corneal edema was graded on a scale of 0 to 4. All eight eyes treated with 0.5% povidone-iodine had no corneal edema, whereas all eight eyes treated with 1.0% showed corneal edema. Although there is little risk of 0.05 mL of povidone-iodine entering the anterior chamber during surgery, the povidone-iodine concentration used for ocular surface irrigation should be set at 0.5% or below for safety purpose.

In a study that evaluated the effect of povidone-iodine on the retina, 0.1 mL of 0.5%, 1%, 2% or 5% povidone-iodine was injected intravitreally into rabbit eyes (*n* = 6 each) [66]. Electroretinograms were recorded and b-wave/a-wave ratios were measured at 1, 7 and 14 days after injection. No significant differences were observed in the 0.5%, 1% and 2% groups compared to the saline-injected group. Pathological examination was performed at 15 days after injection, and the whole circumference of the retina was observed in vertical sections centering on the optic disc. In the group injected with 5% povidone-iodine, extensive inflammatory cell infiltration was found in six of six eyes and retinal detachment in five of six eyes. At 2% povidone-iodine, retinal degeneration and inflammatory cell infiltration were found in two of six eyes. At 1%, localized inflammatory cell infiltration was seen in one of six eyes. At 0.5%, no retinal abnormalities were detected. Although it is unlikely that 0.05 mL of povidone-iodine would enter the vitreous during surgery, these findings confirm that povidone-iodine at concentration of 0.5% or below should be used for ocular surface irrigation.

When the effects of povidone-iodine on corneal epithelial cells, corneal endothelial cells and retina are evaluated comprehensively, povidone-iodine at a concentration of 0.5% or below is recommended for irrigation of the ocular surface. A previous study has also reported 0.05–0.5% as the safe and useful concentration range for the retina [67]. Based on these results, the authors use 0.25% povidone-iodine, which is the median of 0.05 to 0.5%, for ocular surface irrigation during surgery [28]. When the 0.25% povidone-iodine solution applied to the ocular surface is diluted 2.5-fold, the resulting 0.1% povidone-iodine has the highest bactericidal effect, as seen in Figure 6.

### 8.3. Preparation, Color and Storage of 0.25% Povidone-Iodine

In the operating room, 0.25% povidone-iodine is prepared by adding an appropriate volume of 5% or 10% povidone-iodine into a 100-mL or 250-mL physiological saline bottle (Figure 8). Note that the solution should not be prepared in distilled water, because washing the cornea with a solution diluted in distilled water may cause corneal damage due to low osmotic pressure [68]. A washing needle is attached to the cap of the bottle, which will be convenient for irrigating the eye surface. The solution is prepared twice daily, in the morning and afternoon. During surgery, the 0.25% povidone-iodine is dispensed into a cup on the operating table, and a syringe is used for washing.

The color of povidone-iodine reflects the amount of effective iodine [69]. A 5–10% solution is dark brown, a 0.25% solution is brown and a 0.025% solution is light brown. When 0.25% povidone-iodine in an open container is left at room temperature, it decolorizes over time. When 0.25% povidone-iodine in a closed container is placed at room temperature for 60 min, there is no decolorization and no loss of bactericidal effect. When 0.25% povidone-iodine prepared in physiological saline in a closed container is stored in a refrigerator, there is no decolorization even after 1 month.

## 9. Clinical Application of 0.25% Povidone-Iodine

### 9.1. Preoperative Cleaning of Eyelid Skin

Use a sponge soaked in 5% or 10% povidone-iodine to clean the eyelid skin [70]. Note that cotton balls generate fibers, and these fibrous materials may be introduced into the eye by adhering to surgical instruments [71]. Care should be taken to prevent the disinfectant from spreading inside the eyelid, as 5% and 10% povidone-iodine can cause corneal damage. The area that has been cleaned with povidone-iodine is stained yellow, which has the advantage of indicating the applied area. To clean the eyelid skin, use a sponge impregnated with povidone-iodine to draw concentric circles while moving toward the periphery (Figure 9). Repeat this procedure 3–4 times. The area of cleaning should extend to slightly above the eyebrow on the superior side, slightly beyond the bridge of the nose on the medial side, between the lateral canthus and middle of the ear on the lateral side and around the nasal wing on the inferior side. There is also a misunderstanding that 5% or 10% povidone-iodine does not exhibit sufficient bactericidal effect unless the applied solution is dried. What it really means is that an exposure time of 2 to 3 min (approximate time taken for the solution to dry) is needed to achieve sufficient bactericidal effect. After application, the bactericidal effect persists as long as povidone-iodine remains and is not removed by washing with physiological saline. Without removing the applied 5% or 10% povidone-iodine, irrigate the conjunctival sac with 0.25% povidone-iodine. The 0.25% povidone-iodine in the conjunctival sac can be expected to exhibit bactericidal effect if it is not rinsed with saline. The periorbital area stained yellow by 5% or 10% povidone-iodine also serves to identify the eye to be operated on during draping.

### 9.2. Irrigating the Ocular Surface during Cataract Surgery

In cataract surgery, the surgical field should be thoroughly irrigated at the beginning of surgery, before inserting the intraocular lens, and at the end of surgery [28]. “The Shimada technique” [72] refers to the method of repetitively irrigating the ocular surface every 20–30 s during cataract surgery instead of washing with saline (Figure 10). In cataract surgery, the fluid used to irrigate the ocular surface is collected into the fluid catch bag. Bacteria were detected in the fluid catch bag in 23.1% of the eyes when the ocular surface was irrigated with saline [73]. On the other hand, when surgery was performed using 0.25% povidone-iodine to irrigate the ocular surface, the bacterial detection rate in the fluid catch bag was significantly reduced to 3.8% (*p* = 0.0041). Bacteria were still detected in the fluid catch bag occasionally even though 0.25% povidone-iodine was used, probably because the povidone-iodine was diluted by the irrigating fluid [73]. Therefore, care should be taken not to drop unused instruments in the fluid catch bag. In case instruments are dropped accidentally into the bag, they may be washed with 0.25% povidone-iodine before use.

### 9.3. Effectiveness in Cataract Surgery

In a study on the effectiveness of using povidone-iodine during cataract surgery, the ocular surface was irrigated with saline in 200 eyes and with 0.25% povidone-iodine in 200 eyes (Table 3) [28]. In the povidone-iodine group, the ocular surface was washed with 0.25% povidone-iodine every 20–30 s. The bacterial detection rate in the anterior chamber at the end of surgery was significantly lower in eyes washed with 0.25% povidone-iodine (0/200 eyes, 0%) compared to eyes washed with saline (10/200 eyes, 5%) (*p* = 0.0017). The corneal endothelial cell density at day 7 after surgery was not significantly different between the saline (2463 ± 269/mm^2^) and the 0.25% povidone-iodine groups (2338 ± 204/mm^2^) (*p* = 0.4044). Since povidone-iodine has a smaller molecular weight than antibacterial drugs, free iodine readily penetrates the cornea [74,75]. Iodide ions were detected in the anterior chamber at a concentration of 0.0075% at the beginning of cataract surgery and 0.0045% at the end of surgery [28]. The surgical procedures probably resulted in a decrease of povidone-iodine passing into the anterior chamber at the end of surgery. Since the effective concentration of povidone-iodine is 0.005% or higher, the intraocular tissues in the anterior chamber are conceivably also protected by the bactericidal effect.

The current American Academy of Ophthalmology and European Society of Cataract and Refractive Surgeons recommendations regarding povidone-iodine use suggest using 5% povidone-iodine before surgery, as well as an alternative dosing strategy using dilute povidone-iodine repetitively throughout cataract surgery (0.25% every 30 s) [76]. 

### 9.4. Effectiveness in Intravitreal Injection

In intravitreal injection, bacteria are introduced into the vitreous via the needle tip [77]. A study evaluated the effectiveness of preservative-free 0.6% povidone iodine eye drops as perioperative prophylaxis in patients undergoing intravitreal injection randomized to a group receiving 0.6% povidone iodine eye drops for three days before injection, and a control group receiving placebo. Bacterial growth from conjunctival swab cultures was significantly lower after 0.6% povidone iodine prophylaxis compared to baseline and to placebo prophylaxis (*p* < 0.001). However, the bacteria eradication rate in the 0.6% povidone iodine group was 82% and did not achieve 100% [78].

The effect of 0.6% povidone iodine eye drops shows a rapidly bactericidal effect [79,80]. Considering that bacteria are present in the folds of the conjunctiva, povidone iodine irrigation of the ocular surface is more effective than instillation on the conjunctiva [81]. For intravitreal injections, it is better to wash with a few ml of 0.6% povidone iodine shortly before, just before and immediately after the injection. Using 0.25% povidone-iodine to irrigate the conjunctival sac both before and after injection, we reported 0 cases of suspected or proven infectious endophthalmitis in 12,523 intravitreal injections (95% confidence interval 0 to 0.00024%) [82].

## 10. Clinical Application of 0.025–0.1% Povidone-Iodine

It is also known that the incidence and severity of dry eye symptoms may increase after cataract surgery. To achieve the best outcome in cataract surgery, a healthy ocular surface is crucial [83]. In patients who have already developed corneal epithelial cell damage caused by diabetes or dry eye, ocular surface irrigation with 0.25% povidone-iodine may transiently exacerbate the corneal damage. The authors previously showed the usefulness and safety of a method of performing vitrectomy for postoperative endophthalmitis using an irrigating fluid containing 0.025% povidone-iodine [84]. As an application of this method, we investigated the effectiveness of performing cataract surgery while irrigating the ocular surface with 0.025% povidone-iodine. In this study, the ocular surface was irrigated with physiological saline in 200 eyes and with 0.025% povidone-iodine in 100 eyes. The bacteria detection rate in the anterior chamber at the end of surgery was significantly lower when using 0.025% povidone-iodine (0/100, 0%) than with saline (10/200, 5%) (*p* = 0.0340). There was no significant difference in corneal endothelial cell density at day 7 after surgery [85]. This study shows that povidone-iodine even when diluted to a low concentration of 0.025% effectively reduces bacterial contamination in the anterior chamber. When 0.025% povidone-iodine is used to irrigate the ocular surface, it is estimated to be diluted to around 0.01%. As seen in Figure 6, this concentration is within the range that provides high bactericidal effect.

Not only 0.025%, but concentrations of 0.025–0.1% povidone-iodine may be used for ocular surface irrigation in patients who have already developed corneal epithelial damage caused by diabetes or dry eye. This is supported by a study of 90 eyes undergoing cataract surgery, in which the ocular surface was irrigated with 0.05% povidone-iodine for 30 s before surgery. When evaluation was conducted using tear film breakup time, corneal fluorescein staining, lacrimal river height, Schirmer test 1 and conjunctival bacteria detection as indicators, 0.05% povidone-iodine exhibited bactericidal effect and was beneficial in the recovery of ocular surface function [86].

## 11. Pre-Cataract Surgery Povidone-Iodine Instillation/Irrigation and Endophthalmitis

A multicenter, nonrandomized, prospective, controlled study evaluated 0.66% povidone-iodine eye drops (IODIM^®^) as perioperative prophylaxis against the conjunctival bacterial flora of patients who underwent cataract surgery and found that 0.66% povidone-iodine eye drops used for three days prior to cataract surgery effectively reduced the conjunctival bacterial load. The 0.66% povidone-iodine eye drops may represent a valid perioperative prophylactic antiseptic adjuvant treatment to protect the ocular surface from microbial contamination in preparation of the surgical procedure [87].

There are no reports on the frequency of postoperative endophthalmitis after cataract surgery with repetitive ocular surface irrigation with povidone-iodine during surgery. One study analyzed the association of the incidence of endophthalmitis after cataract surgery with the changes of preoperative prophylaxis regimen over a 20-year period in one institution, and reported the usefulness of povidone-iodine instillation/irrigation at the beginning of surgery [88]. The incidence of endophthalmitis was 0.291% during the period 1990–1992 when no prophylactic regimen against infectious diseases was used, 0.170% during the period 1993–1998 using the regimen of preoperative antibiotic eye drops + preoperative 10% povidone-iodine around the eye + 1 drop of 1% povidone-iodine in the conjunctival sac at the start of surgery and 0.065% during the period 1999–2009 using the regimen of preoperative antibiotic eye drops + preoperative 10% povidone-iodine around the eye + single irrigation of the conjunctival sac with 10 mL of 1% povidone-iodine, with significant differences between the three groups (*p* < 0.001).

## 12. Intracameral Antibiotics as Prophylaxis of Postoperative Endophthalmitis

A prospective randomized partially masked multicenter cataract surgery study in nine European countries which recruited 16,603 patients concluded that use of intracameral cefuroxime at the standard dose of 1 mg/0.1 mL at the end of surgery reduced the incidence of postoperative endophthalmitis [37]. Furthermore, a meta-analysis reported that intracameral cefuroxime and moxifloxacin at standard doses reduced endophthalmitis rates compared to controls with minimal or no toxicity events [38]. However, use of vancomycin has been reported to be associated with hemorrhagic occlusive retinal vasculitis [89] and postoperative endophthalmitis due to cefuroxime-resistant strains despite intracameral antibiotic prophylaxis is an issue [36,90]. Combined use of standard concentrations of intracameral antibiotics and 0.25% povidone-iodine ocular surface irrigation may be a strategy to be explored. 

## 13. Iodine Hypersensitivity

There is no report of anaphylaxis caused by the use of povidone-iodine in the field of ophthalmology [91]. There is a case of a 9-year-old boy who developed anaphylaxis after using povidone-iodine to disinfect the skin [92]. In another case report, a 59-year-old woman who had had several episodes of contact urticaria after hair treatment developed anaphylaxis after vaginal application of povidone-iodine solution for disinfection [93].

## 14. Conclusions

Povidone-iodine has several advantages including established effective and safe concentrations for ocular tissues, broad-spectrum antimicrobial activity, short exposure time for microbial killing, the absence of resistant strains, low cost and ease of use worldwide. Repetitive washing of the ocular surface with 0.25% povidone-iodine every 20–30 s throughout cataract surgery is expected to be a preventive method for endophthalmitis. 

## Figures and Tables

**Figure 1 jcm-10-03611-f001:**
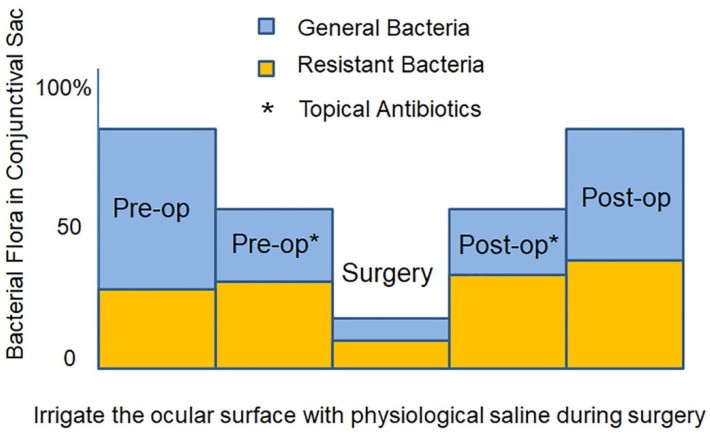
Using antibiotic eye drops before and after surgery and washing with saline during surgery. Preoperative use of antibiotic eye drops not only has limited disinfection effect, but the procedure also increases the proportion of resistant bacteria after surgery. Washing with saline during surgery only reduces the number of bacteria on the ocular surface, and the risk of bacteria entering the eye remains. (Authors’ unpublished images).

**Figure 2 jcm-10-03611-f002:**
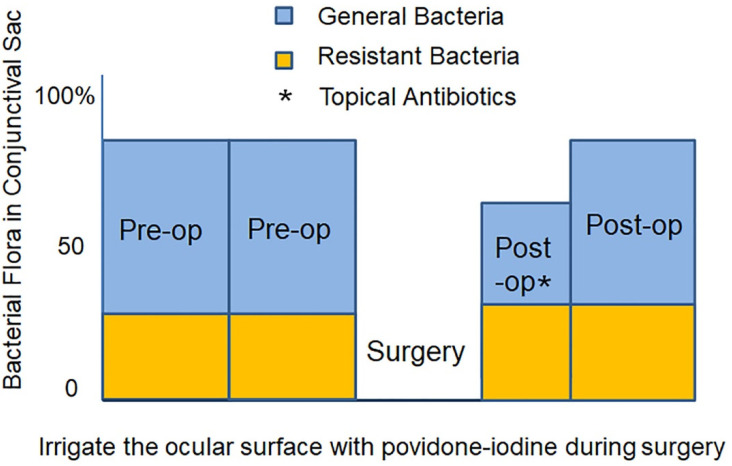
Discontinuing preoperative antibiotic eye drops, performing intraoperative povidone-iodine irrigation and shortening the period of postoperative antibiotic eye drops. By attempting to render the ocular surface microbe-free by irrigation with povidone-iodine, the risk of bacteria entering the eye is reduced. By stopping preoperative use of antibiotic eye drops and shortening the period of using postoperative antibiotic eye drops, the increase in resistant bacteria is suppressed. (Authors’ unpublished images).

**Figure 3 jcm-10-03611-f003:**
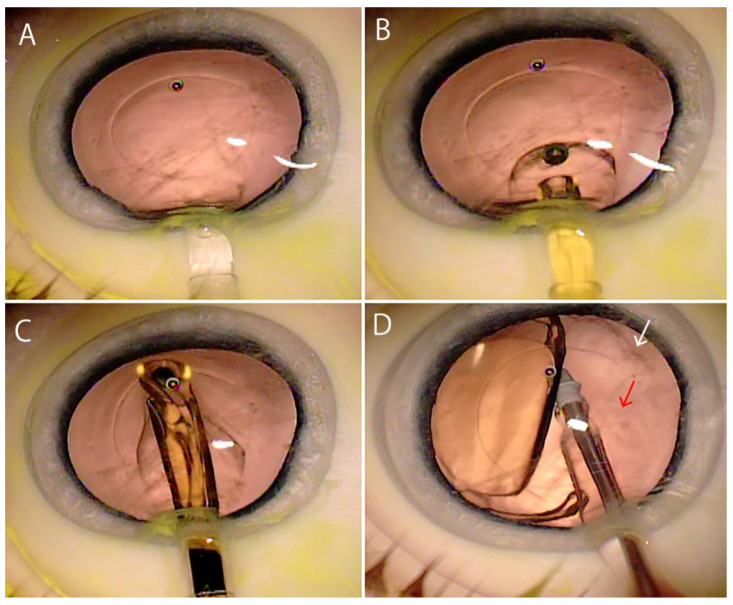
Application of fluoresbrite carboxylate microspheres (1 µm) onto the ocular surface of porcine eye with eyelids. (**A**) Accompanying insertion of the intraocular lens, (**B**) the fluoresbrite microspheres are introduced into the anterior chamber in the shape of balloon due to displacement by the ophthalmic viscosurgical device. (**C**) The fluoresbrite microspheres spread around the intraocular lens. (**D**) Even after washing the anterior and posterior surfaces of the intraocular lens, the fluoresbrite microspheres remain in the anterior chamber (white arrow) and a small amount of fluoresbrite microspheres enter the vitreous body (red arrow). (Authors’ unpublished images).

**Figure 4 jcm-10-03611-f004:**
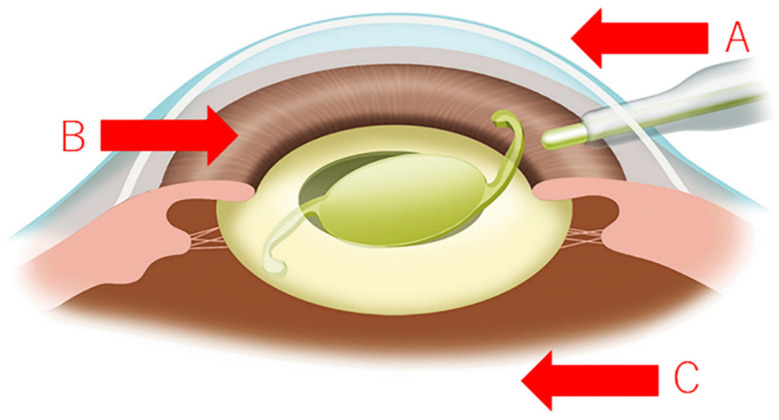
Prevention of endophthalmitis after cataract surgery (three-step approach); (**A**) The most important measure is “border control” to prevent bacteria, fungi, viruses and other microorganisms from entering the eye. (**B**) The second measure is to thoroughly wash out the bacteria that have entered the anterior chamber. (**C**) For the third measure, bacteria that remain in the anterior chamber and vitreous are eliminated by antibacterial drugs. As a “border control” measure, it is useful to irrigate the ocular surface with 0.25% povidone-iodine every 20–30 s during cataract surgery to transiently sterilize the ocular surface instead of washing with physiological saline. (Authors’ unpublished images).

**Figure 5 jcm-10-03611-f005:**
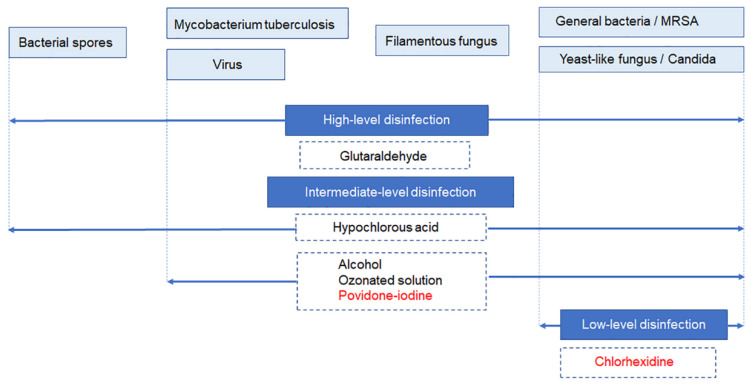
Three levels of disinfectants classified according to features including difference in antimicrobial spectrum. High-level disinfectants are used for disinfecting instruments. Among the intermediate-level disinfectants, only povidone-iodine has the established safe concentration for ocular tissues. Among the intermediate-level agents, chlorhexidine is used in intravitreal injection. However, since the safe concentration for intraocular tissues has not been established, chlorhexidine cannot be used for intraoperative irrigation of the ocular surface. (Authors’ unpublished image).

**Figure 6 jcm-10-03611-f006:**
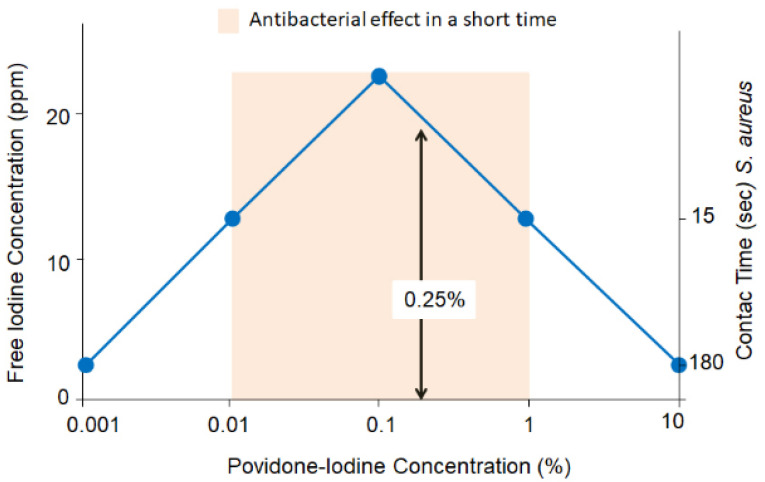
Concentrations of povidone-iodine, effective iodine concentration and exposure time required to obtain microbicidal effect. 0.1% povidone-iodine has free iodine concentration of 24 ppm with the greatest bactericidal effect, and the time required for microbial killing is as short as 15 s. When used to irrigate the ocular surface, 0.25% povidone-iodine is diluted approximately 2.5-fold to 0.1%. (Authors’ unpublished images).

**Figure 7 jcm-10-03611-f007:**
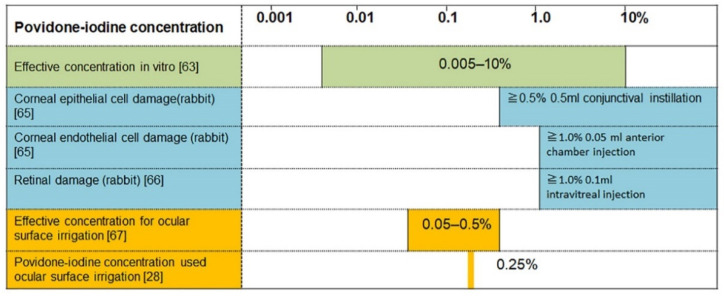
Concentrations of povidone-iodine used for ocular surface irrigation. The concentrations that cause damage to corneal endothelial cells, corneal epithelial cells and retina are shown; 0.025% povidone-iodine is the median for the concentration range of 0.05–0.5% that is suitable for ocular surface irrigation. (Modified from [28]).

**Figure 8 jcm-10-03611-f008:**
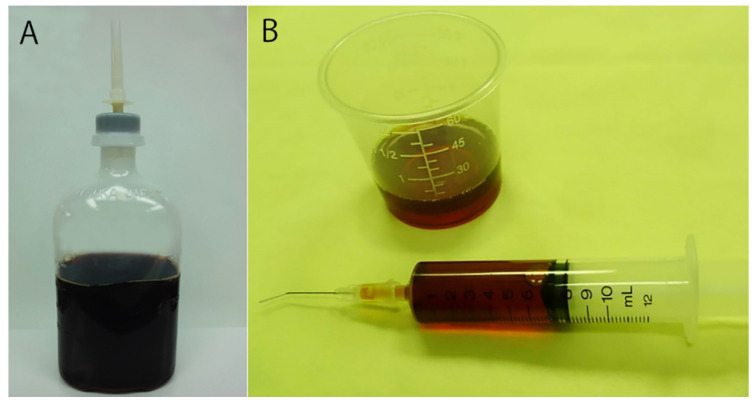
Preparation and use of 0.25% povidone-iodine. (**A**) Inject an appropriate volume of 5% or 10% povidone-iodine into a 250-mL saline bottle to prepare a 0.25% solution, and attach a cleaning needle to the cap. This can also be used to clean the ocular surface before surgery. (**B**) Dispense the solution from the bottle into a cup placed on the operating table, put a syringe in the cup and use during cataract surgery. (Modified from [69]).

**Figure 9 jcm-10-03611-f009:**
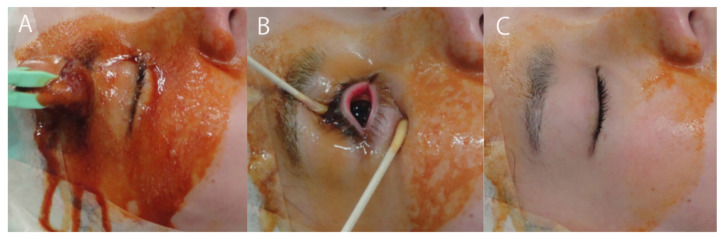
Preoperative cleaning of the eyelid skin. (**A**) Disinfect the skin around the eye with 10% povidone-iodine. (**B**) Disinfect the ocular surface with 0.25% povidone-iodine. To maintain the bactericidal effect, do not rinse off the 0.25% povidone-iodine with saline. (**C**) Wipe off the 10% povidone-iodine around the eye, but leave the peripheral povidone-iodine which helps to identify the eye to be operated on. (Modified from [69]).

**Figure 10 jcm-10-03611-f010:**
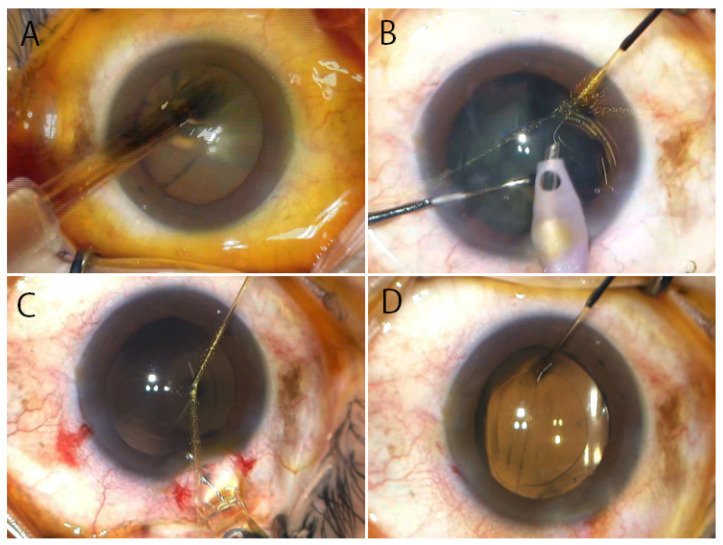
Irrigation of ocular surface with 0.25% povidone-iodine during cataract surgery instead of washing with saline (The Shimada technique). (**A**) Irrigate the ocular surface with 0.25% povidone-iodine after placing the speculum. (**B**) While irrigating the cornea, make sure that 0.25% povidone-iodine accumulates in the conjunctival sac. (**C**) Since 0.25% povidone-iodine shows a bactericidal effect within 15 s, irrigate every 20–30 s during surgery; especially, irrigate before inserting the intraocular lens. (**D**) Irrigate also at the end of surgery. The applied 0.25% povidone-iodine flows from the cornea to the conjunctiva, so the visibility of the anterior chamber is good. Fresh 0.25% povidone-iodine is applied to the cornea before the corneal surface becomes dry due to flow of 0.25% povidone-iodine to the conjunctiva. (Authors’ unpublished images).

**Table 1 jcm-10-03611-t001:** Antimicrobial characteristics of antimicrobial agents and povidone-iodine.

	Mechanism	Selectivity	Tolerance	Contact Time
Antimicrobial Agent	Cell wall/protein/DNA synthesis inhibition	+	+	15–140 min
Povidone-Iodine	Membrane protein denaturation	−	−	15–180 s

DNA: deoxyribonucleic acid, min: minutes, s: seconds.

**Table 2 jcm-10-03611-t002:** Microbicidal features and sites and methods of use for high and low concentrations of povidone-iodine.

	Povidone-Iodine 2.5–10%	Povidone-Iodine 0.05–0.5%
Free iodine concentration (ppm)	3–10	15–24
Exposure time (s)	30–180	15
Microbicidal duration	long	short
Site of use	Skin cleansing	Ocular surface irrigation
Method of use	Once cleaning	Repeatedly irrigated

ppm: parts per million.

**Table 3 jcm-10-03611-t003:** Bacterial contamination rate, and corneal endothelial cell density when 0.25% povidone-iodine was used to repetitively irrigate the ocular surface during cataract surgery.

Method of Ocular Surface Irrigation (No.)	Microbial Contamination Rate (%)		Corneal Endothelial Cell Density (/mm^2^)	
	Start of Surgery Ocular Surface Fluid	End of Surgery Anterior Chamber Fluid	Preop	Day 7 Postop
Physiological saline (*n* = 200)	11/200 (5.5%) *	10/200 (5.0%) **	2614 ± 233 ^+^	2463 ± 269 ^++^
	CNS (7) *Micrococcus* sp. (1) *Enterococcus faecalis* (1) *Staphylococcus aureus* (1) *Corynebacterium* spp. (1)	CNS (6) *Enterococcus* sp. (1) *Enterococcus faecalis* (1) *Staphylococcus aureus* (1) *Klebsiella pneumoniae* (1)		
0.25% povidone-iodine (*n* = 200)	12/200 (6.0%) *	0/200 (0%) **	2534 ± 173 ^+^	2338 ± 204 ^++^
	CNS (8) *Staphylococcus aureus* (2) *Micrococcus* sp. (1) *Klebsiella* spp. (1)			
*p*	>0.99 *	0.0017 **	0.2254 ^+^	0.4044 ^++^

CNS = coagulase-negative *Staphylococcus* sp. *, **: Fisher exact probability test. ^+^, ^++^: Mann–Whitney. Data extracted from [28].

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
