# Peer review of "Cataract Surgery by Intraoperative Surface Irrigation with 0.25% Povidone–Iodine"

_jcm, 2021, doi:10.3390/jcm10163611_

Round 1

Reviewer 1 Report

The authors present a manuscript addressing the use of 0.25% povidone-iodine for surface irrigation during cataract surgery. Following a short overview on current literature and practice, the authors present different approaches for endophthalmitis prevention. The authors conclude that repeated application of Povidone-Iodine has several advantages in endophthalmitis prophylaxis. The manuscript is well-written and structured, but I have serious concerns regarding the premise of the discussion, as important literature, essential safety information and sufficient evidence for the suggestions are missing.

Minor comments:

  • Line 27-34: In my personal opinion, the comparison to CoVid-19 pandemic as an opening statement is misleading in the context of the manuscript.
  • Line 55: I suggest using another word than “Despite”, as there is no correlation between incidence and prognosis of postoperative endophthalmitis.
  • Reference [13] is probably wrong: Jpn J Ophthalmol Surg
  • 1/2: What does the blue boxes indicate? What does the term “healthy” describe in this context?
  • Line 115: I suggest adding an “A” at the beginning of the sentence.
  • Figure 3: Picture E is missing.
  • Line 125: I suggest adding an “a” before balloon.
  • In Fig. 10, I do not think that the rinsing of the ocular surface - as described in the image description - is well illustrated. Rather, you can see the individual steps of the surgery where dripping is recommended. I recommend changing the image description.
  • Line 179-183: According to current literature, this statement is probably misleading.
  • Line 504-513 should be moved from the “Conclusio” section.
  • Citation [57] is the same as citation [16].

Major comments:

  • Line 78: I think the definition for infection prophylaxis is too stringent as there are more ways of infection prophylaxis than antibiotic eye drops 4 times a day from 1-3 days before surgery. Currently, the state of the art prefers intracameral antibiosis, not pre- or postoperative antibiosis. The comparison to an obsolete premise should be removed by discussing current evidence (especially intracameral antibiosis).
  • I have major concerns regarding the manuscript as the authors suggest new methods and spare out key studies, for example by ESCRS in 2007 (see below), and discussion of current state of the art, namely intracameral antibiosis.
  • Besides incomplete discussion of current state of the art, there is a lack of randomized controlled trials (RCT) regarding the endophthalmitis rate by the proposed method of the authors, as stated in paragraph 10 of the manuscript. In my opinion, a method should not be suggested or proposed if there is no evidence by RCTs. Irrigation by 0.25% PI every 20-30 seconds may be a promising approach, but currently there is no evidence for recommendation.
  • Line 195-196: “As a “border control” measure, it is useful to irrigate the ocular surface with 0.25% povidone-iodine every 20-30 seconds during cataract surgery to transiently sterilize the ocular surface.” – By rinsing every 20-30 seconds, how can it be guaranteed that PI has enough application time?
  • Furthermore, can it be guaranteed, that the surgeon has enough sight during the surgery, when PI is used?
  • Line 56: Reference [8] (Nowak et al 2019) does not indicate endophthalmitis rate. Furthermore, as endophthalmitis rate is directly dependent on prophylaxis regime, important key studies are missing, for example:
    • ESCRS Endophthalmitis Study Group Prophylaxis of postoperative endophthalmitis following cataract surgery: Results of the ESCRS multicenter study and identification of risk factors, Journal of Cataract & Refractive Surgery: June 2007 - Volume 33 - Issue 6 - p 978-988 doi: 10.1016/j.jcrs.2007.02.032;
    • Aravind Haripriya, David F. Chang, Sathvik Namburar, Anand Smita, Ravilla D. Ravindran, Efficacy of Intracameral Moxifloxacin Endophthalmitis Prophylaxis at Aravind Eye Hospital, Ophthalmology, Volume 123, Issue 2, 2016, Pages 302-308, https://doi.org/10.1016/j.ophtha.2015.09.037.
    • Bowen RC, Zhou AX, Bondalapati S, et al Comparative analysis of the safety and efficacy of intracameral cefuroxime, moxifloxacin and vancomycin at the end of cataract surgery: a meta-analysis. British Journal of Ophthalmology 2018;102:1268-1276.

Author Response

Comments and Suggestions for Authors

The authors present a manuscript addressing the use of 0.25% povidone-iodine for surface irrigation during cataract surgery. Following a short overview on current literature and practice, the authors present different approaches for endophthalmitis prevention. The authors conclude that repeated application of Povidone-Iodine has several advantages in endophthalmitis prophylaxis. The manuscript is well-written and structured, but I have serious concerns regarding the premise of the discussion, as important literature, essential safety information and sufficient evidence for the suggestions are missing.

Minor comments:

Line 27-34: In my personal opinion, the comparison to CoVid-19 pandemic as an opening statement is misleading in the context of the manuscript.

Response: We have rewritten Introduction to provide a short summary regarding the topic and the main aim of the study, and moved the sentences about COVID-19 to the new section 2. Border control.

Line 55: I suggest using another word than “Despite”, as there is no correlation between incidence and prognosis of postoperative endophthalmitis.

Response: The sentence has been changed as follows:

Lines 78-79: “Although the incidence of endophthalmitis after cataract surgery is low with reports of 0.076% [10], 0.02% [11] and 0.048% [12], the prognosis is unfavorable.”

Reference [13] is probably wrong: Jpn J Ophthalmol Surg

Response: We apologize for a mistake in the name of the journal. It should read Jpn. J. Ophthalmic Surg. The reference has been corrected as follows:

Reference 16: “Asari, S. Post-cataract surgery endophthalmitis. Jpn. J. Ophthalmic. Surg. 2006,19,365-370. [Japanese] 

1/2: What does the blue boxes indicate? What does the term “healthy” describe in this context?

Response: We have changed “Healthy” to “Pre-op” or “Post-op”. Figures 1 and 2 have been changed.

Line 115: I suggest adding an “A” at the beginning of the sentence.

Response: We have added “A” at the beginning of the sentence.

Line 141: “A study using molecular epidemiology…….”

Figure 3: Picture E is missing.

Response: We apologize for the mistake. (E) has been deleted.

Figure 3 legend: “…… and (E) a small amount of fluoresbrite….”

Line 125: I suggest adding an “a” before balloon.

Response: We have inserted “a” before balloon

Line 151: ……in the shape of a balloon..

In Fig. 10, I do not think that the rinsing of the ocular surface - as described in the image description - is well illustrated. Rather, you can see the individual steps of the surgery where dripping is recommended. I recommend changing the image description.

Response: The original reference used the term “irrigation” (Shimada H et al. Reduction of anterior chamber contamination rate after cataract surgery by intraoperative irrigation with 0.25% povidone-iodine. Am. J. Ophthalmol. 2011,151(1),11-17). We have changed Figure 10 (D) to an image showing irrigation.

Line 179-183: According to current literature, this statement is probably misleading.

Response: We have deleted the part pointed out by the reviewer:

Since a portion of the antibiotics injected into the anterior chamber passes into the vitreous body, intracameral antibiotics can be expected to be effective also against bacteria that have entered the vitreous. Despite intracameral injection of antibiotics, this measure will not be effective if resistant bacteria have invaded the anterior chamber and vitreous. Postoperative antibiotic eye drops sufficiently penetrate the anterior chamber and can be expected to be effective against bacteria remaining in the anterior chamber [31].

Line 504-513 should be moved from the “Conclusion” section.

Response: Following the suggestion of another reviewer, we have deleted 12. Conclusion.

Povidone-iodine has several advantages including established effective and safe concentrations for ocular tissues, broad-spectrum antimicrobial activity, short exposure time for microbial killing, absence of resistant strains, low cost, and easy to use worldwide.

We propose that a three-step approach is important for the prevention of endophthalmitis after cataract surgery. First, the most important measure is to implement “border control” to prevent microorganisms from entering the eye, including not only bacteria but also fungi and viruses. The second measure is thorough washing of the anterior chamber. The third measure is to is to eliminate the bacteria remaining in the anterior chamber and vitreous by antibacterial drugs. As a “border control” measure, we propose a method of irrigating the ocular surface with 0.25% povidone-iodine every 20–30 seconds during cataract surgery to transiently render the ocular surface almost microbe-free. Ocular surface irrigation with 0.025-0.1% povidone-iodine may be useful for patients who have already developed corneal epithelial damage caused by diabetes or dry eye. When combined with intracameral antibiotic injection, these regimens can be expected to further reduce the frequency of endophthalmitis.

It should be noted that for research on povidone-iodine, a small bacterial inoculum should be used for studies that are based on the premise of bactericidal effect on the ocular surface. When a larger number of bacteria is used, although one-time application of povidone-iodine solution at low concentration requires short exposure time, the bactericidal effect does not last, and as a result may be evaluated as ineffective. Conversely, for one-time application of high concentration of povidone-iodine, although microbial killing requires long exposure, the long-acting bactericidal effect may lead to evaluation of high efficacy. One should avoid the misleading conclusion that one-time application of low concentration povidone-iodine is ineffective for disinfection of ocular surface by designing an experiment using a skin model with large bacterial load.

We have rewritten shortened Conclusion to include statements supported by clinical evidence as follows:

“Povidone-iodine has several advantages including established effective and safe concentrations for ocular tissues, broad-spectrum antimicrobial activity, short exposure time for microbial killing, absence of resistant strains, low cost, and easy to use worldwide. Using 5-10% povidone-iodine for skin disinfection before surgery and repetitive washing of the ocular surface with 0.25% povidone-iodine every 30 seconds throughout cataract surgery effectively reduce the risk of postoperative endophthalmitis.”

Citation [57] is the same as citation [16].

Response: We apologize for the duplication. We have deleted [57].

Shimada, H.; Arai, S.; Nakashizuka, H.; Hattori, T.; Yuzawa, M. Reduction of anterior chamber contamination rate after cataract surgery by intraoperative irrigation with 0.25% povidone-iodine. Am. J. Ophthalmol. 2011,151(1),11-17.

Major comments:

Line 78: I think the definition for infection prophylaxis is too stringent as there are more ways of infection prophylaxis than antibiotic eye drops 4 times a day from 1-3 days before surgery. Currently, the state of the art prefers intracameral antibiosis, not pre- or postoperative antibiosis. The comparison to an obsolete premise should be removed by discussing current evidence (especially intracameral antibiosis).

Response: We have revised the sentence and mentioned intracameral antibiotics as a method of infection prophylaxis. However, instillation of antibiotic eye drops 4 times a day from 1–3 days before surgery remains the mainstay of infection prophylaxis before cataract surgery. We have revised this part as follows:

Lines 100-104: Many different practice patterns of infection prophylaxis have been reported around the world [17, 18]. Especially, use of prophylactic intracameral antibiotics varies widely among countries and facilities. In Japan, instillation of antibiotic eye drops 4 times a day from 1–3 days before surgery remains the mainstay of preoperative disinfection before cataract surgery [19].

I have major concerns regarding the manuscript as the authors suggest new methods and spare out key studies, for example by ESCRS in 2007 (see below), and discussion of current state of the art, namely intracameral antibiosis.

We have cited the following two papers.

  1. Endophthalmitis Study Group, European Society of Cataract & Refractive Surgeons. Prophylaxis of postoperative endophthalmitis following cataract surgery: results of the ESCRS multicenter study and identification of risk factors. J. Cataract. Refract. Surg.

2007,33(6),978-988.

  1. Bowen, R.C.; Zhou, A.X.; Bondalapati. S.; Lawyer, T.W.; Snow, K.B.; Evans, P.R.; Bardsley, T.; McFarland, M.; Kliethermes, M.; Shi, D.; Mamalis, C.A.; Greene, T.; Rudnisky, C.J.; Ambati, B.K. Comparative analysis of the safety and efficacy of intracameral cefuroxime, moxifloxacin and vancomycin at the end of cataract surgery: a meta-analysis. Br. J. Ophthalmol. 2018,102(9),1268-1276.

Besides incomplete discussion of current state of the art, there is a lack of randomized controlled trials (RCT) regarding the endophthalmitis rate by the proposed method of the authors, as stated in paragraph 10 of the manuscript. In my opinion, a method should not be suggested or proposed if there is no evidence by RCTs. Irrigation by 0.25% PI every 20-30 seconds may be a promising approach, but currently there is no evidence for recommendation.

Response: Regarding this point, we have deleted 12. Conclusion, according to the comment of another reviewer.

Response: We agree that there is a lack of randomized controlled trials on the incidence of endophthalmitis of our proposed method. However, American Academy of Ophthalmology and European Society of Cataract and Refractive Surgeons recommendations regarding povidone-iodine use suggest using 5% povidone-iodine before surgery, as well as an alternative dosing strategy using dilute povidone-iodine repetitively throughout cataract surgery (0.25% every 30 seconds) [Koerner et al. Povidone-iodine concentration and dosing in cataract surgery. Surv. Ophthalmol. 2018,63(6),862-868.]. Together with plenty of non-RCT clinical evidence indicating the usefulness of this approach, we consider that inclusion of this statement in Conclusion may be permissible. We have shortened the conclusion as follows:

“Povidone-iodine has several advantages including established effective and safe concentrations for ocular tissues, broad-spectrum antimicrobial activity, short exposure time for microbial killing, absence of resistant strains, low cost, and easy to use worldwide. Using 5-10% povidone-iodine for skin disinfection before surgery and repetitive washing of the ocular surface with 0.25% povidone-iodine every 30 seconds throughout cataract surgery effectively reduce the risk of postoperative endophthalmitis.”

Line 195-196: “As a “border control” measure, it is useful to irrigate the ocular surface with 0.25% povidone-iodine every 20-30 seconds during cataract surgery to transiently sterilize the ocular surface.” – By rinsing every 20-30 seconds, how can it be guaranteed that PI has enough application time?

Response: In general, physiological saline is used to wash the ocular surface every 20-30 seconds. Instead of physiological saline, we washed the ocular surface with 0.25% povidone-iodine every 20-30 seconds. We have revised this sentence as follows:

Line 222-224: As a “border control” measure, it is useful to irrigate the ocular surface with 0.25% povidone-iodine every 20-30 seconds during cataract surgery to transiently sterilize the ocular surface instead of washing with physiological saline.

  1. 21. Shimada, H.; Arai, S.; Nakashizuka, H.; Hattori, T.; Yuzawa, M. Reduction of anterior chamber contamination rate after cataract surgery by intraoperative irrigation with 0.25% povidone-iodine. J. Ophthalmol. 2011,151(1),11-17.

Furthermore, can it be guaranteed, that the surgeon has enough sight during the surgery, when PI is used?

Response: We have added the following in the legend of Figure 10.

“The applied 0.25% povidone-iodine flows from the cornea to the conjunctiva, so the visibility of the anterior chamber is good.”

Line 56: Reference [8] (Nowak et al 2019) does not indicate endophthalmitis rate. Furthermore, as endophthalmitis rate is directly dependent on prophylaxis regime, important key studies are missing, for example:

Response: We have changed Nowark et al. 2019 to Haripriya et al. 2016 as suggested by the reviewer. The sentence has been changed to the following:

Lines 79-81: “Although the incidence of endophthalmitis after cataract surgery is low with reports of 0.076% [10], 0.02% [11] and 0.048% [12], the prognosis is unfavorable.”

[11] “Haripriya, A.; Chang, D.F.; Namburar, S.; Smita, A.; Ravindran, R.D. Efficacy of intracameral moxifloxacin endophthalmitis prophylaxis at Aravind Eye Hospital. Ophthalmology. 2016,123(2):302-308.”

Reviewer 2 Report

If the paper is a review, authors should limit their work to the description of the available literature. No conclusions or suggested approach can be reported in absence of any original research.

Please remove all the introduction from line 27 to line 53. Introduction needs to provide a short summary regarding the topic and justify the main aims of the study. Please also remove subheadings in the introduction. The entire flow of the manuscript need to be improved as well.

Line 55: Please provide a more complete definition of endophthalmitis. Endophthalmitis is the most serious and devastating condition within the eye globe, which may lead to irreversible blindness in the affected eye. It is a purulent inflammationn of the intraocular fluids, i.e. the vitreous and the aqueous humour. Although the aetiology of this condition can be both endogenous and exogenous, it is mostly secondary to intraocular surgeries, injections, or penetrating ocular trauma. Please also mention in a sentence the treatment of this serios complication to provide to readers a more complete overview of the topic.

Please use the following reference:

Urgent Vitrectomy with Vancomycin Infusion, Silicone Oil Endotamponade, and General Antibiotic Treatment in Multiple Cases of Endophthalmitis from a Single Day of Intravitreal Injections-Case Series. J Clin Med. 2021 Mar 4;10(5):1059. doi: 10.3390/jcm10051059. PMID: 33806541; PMCID: PMC7961493.

Authors should underline more in detail that cataract surgery is the most prevalent surgery performed in ophthalmology. With a global prevalence of about 50% in adults over 50 years of age, cataract surgery is the most common operative procedure performed in developed countries.12 In 2010, of the total reported 32.4 million blind and 191 million vision-impaired cases, 10.8 million people were blind and 35.1 million were visually impaired due to cataract.

Please use the following reference:

Early impact of COVID-19 outbreak on eye care: Insights from EUROCOVCAT group. Eur J Ophthalmol. 2021 Jan;31(1):5-9. doi: 10.1177/1120672120960339. Epub 2020 Sep 24. PMID: 32967466.

Rethinking Elective Cataract Surgery Diagnostics, Assessments, and Tools after the COVID-19 Pandemic Experience and Beyond: Insights from the EUROCOVCAT Group. Diagnostics (Basel). 2020 Dec 2;10(12):1035. doi: 10.3390/diagnostics10121035. PMID: 33276612; PMCID: PMC7761628.

All ocular surgeries should be also considered as an ocular surface damaging event. Indded, it is also known that the incidence and severity of dry eye symptoms may increase after cataract surgery. To achieve the best outcome in cataract surgery, a healthy ocular surface is crucial. Patients with more severe ocular surface disease are at higher risk of post-operative complications such as secondary infections.

Please use the following reference:

Prevalence of Ocular Demodicosis and Ocular Surface Conditions in Patients Selected for Cataract Surgery. J Clin Med. 2020 Sep 23;9(10):3069. doi: 10.3390/jcm9103069. PMID: 32977656; PMCID: PMC7598293.

However, povidone iodine antisepsis does not reduce to nil the risk of endophthalmitis after intravitreal therapy, since its incidence after PI application ranges from to 0.02% to 0.3% and a cumulative rate throughout the treatment series was reported in up to 1% of patients.Studies on conjunctival swab after PI antisepsis showed a significant reduction of bacterial load on the eye surface, but not a complete eradication, with a lowest rate of culture-positive swabs of 3%. Likewise, needles used for intravitreal injections, after povidone iodine antisepsis, have been found to be contaminated by bacteria, with a range varying from 0.4% to 21%.

Please discuss more in details these aspects using the following reference:

The Effectiveness of 0.6% Povidone Iodine Eye Drops in Reducing the Conjunctival Bacterial Load and Needle Contamination in Patients Undergoing Anti-VEGF Intravitreal Injection: A Prospective, Randomized Study. J Clin Med. 2019 Jul 13;8(7):1031. doi: 10.3390/jcm8071031. PMID: 31337003; PMCID: PMC6678890.

Line 67-68: Therefore, it is necessary to prevent endophthalmitis using drugs or other agents with a wide antimicrobial spectrum against also multidrug-resistant bacteria and fungi.

Paduch R, Urbanik-Sypniewska T, Kutkowska J, ChorÄ…giewicz T, Matysik-Woźniak A, Zweifel S, Czarnek-Chudzik A, ZaÅ‚uska W, Rejdak R, Toro MD. Ozone-Based Eye Drops Activity on Ocular Epithelial Cells and Potential Pathogens Infecting the Front of the Eye. Antioxidants. 2021; 10(6):968. https://doi.org/10.3390/antiox10060968.                                     

“Continuous use of the same antibiotic eye drops induces growth of bacterial species re-92 sistant to the eye drops by the microbial substitution phenomenon. When administration 93 of postoperative antibiotic eye drops is finished, instead of recovery of the bacterial flora 94 to the preoperative state, there is an increase of resistant bacteria. 95”. Please add a reference.

All Figures need a title and a more clear explanation of what is shown in the figures.

Line 103-105: “Since there 103 is still a risk that a minute number of bacteria may remain in the anterior chamber at the 104 end of surgery, postoperative antibiotic eye drops are considered necessary.” Are you sure? Please add a reference.

The paper needs a professional English Editing and the entire flow needs to be improved.

Author Response

Comments and Suggestions for Authors

If the paper is a review, authors should limit their work to the description of the available literature. No conclusions or suggested approach can be reported in absence of any original research.

Response: We agree with the reviewer’s comment. We have deleted 12. Conclusion.

Povidone-iodine has several advantages including established effective and safe concentrations for ocular tissues, broad-spectrum antimicrobial activity, short exposure time for microbial killing, absence of resistant strains, low cost, and easy to use worldwide.

We propose that a three-step approach is important for the prevention of endophthalmitis after cataract surgery. First, the most important measure is to implement “border control” to prevent microorganisms from entering the eye, including not only bacteria but also fungi and viruses. The second measure is thorough washing of the anterior chamber. The third measure is to is to eliminate the bacteria remaining in the anterior chamber and vitreous by antibacterial drugs. As a “border control” measure, we propose a method of irrigating the ocular surface with 0.25% povidone-iodine every 20–30 seconds during cataract surgery to transiently render the ocular surface almost microbe-free. Ocular surface irrigation with 0.025-0.1% povidone-iodine may be useful for patients who have already developed corneal epithelial damage caused by diabetes or dry eye. When combined with intracameral antibiotic injection, these regimens can be expected to further reduce the frequency of endophthalmitis.

It should be noted that for research on povidone-iodine, a small bacterial inoculum should be used for studies that are based on the premise of bactericidal effect on the ocular surface. When a larger number of bacteria is used, although one-time application of povidone-iodine solution at low concentration requires short exposure time, the bactericidal effect does not last, and as a result may be evaluated as ineffective. Conversely, for one-time application of high concentration of povidone-iodine, although microbial killing requires long exposure, the long-acting bactericidal effect may lead to evaluation of high efficacy. One should avoid the misleading conclusion that one-time application of low concentration povidone-iodine is ineffective for disinfection of ocular surface by designing an experiment using a skin model with large bacterial load.

Response: We have shortened Conclusion to include only statements supported by clinical evidence as follows:

“Povidone-iodine has several advantages including established effective and safe concentrations for ocular tissues, broad-spectrum antimicrobial activity, short exposure time for microbial killing, absence of resistant strains, low cost, and easy to use worldwide. Using 5-10% povidone-iodine for skin disinfection before surgery and repetitive washing of the ocular surface with 0.25% povidone-iodine every 30 seconds throughout cataract surgery effectively reduce the risk of postoperative endophthalmitis.”

Please remove all the introduction from line 27 to line 53. Introduction needs to provide a short summary regarding the topic and justify the main aims of the study. Please also remove subheadings in the introduction. The entire flow of the manuscript need to be improved as well.

Response: We have rewritten Introduction to provide a summary of the topic and the main aims of the paper, as follows (Lines 27-43):

“1. Introduction

   Cataract surgery is the most common surgery performed in ophthalmology. With a global cataract prevalence of approximately 50% among adults over the age of 50, cataract surgery is the most common surgical procedure performed in developed countries. Of the overall 32.4 million blind and 191 million visually impaired reported in 2010, 10.8 million people were blind and 35.1 million were visually impaired due to cataract [1]. Among the complications of cataract surgery, postoperative endophthalmitis is the most serious condition, and prevention of postoperative endophthalmitis is an important issue. Postoperative endophthalmitis after cataract surgery is typically caused by the patient's own conjunctival normal bacterial flora. The current problem is an increase of multidrug-resistant bacteria and fungi causing endophthalmitis. In this paper, we introduce a "3-step approach" to prevent post-cataract endophthalmitis caused by intraocular migration of bacteria during cataract surgery. Among the "3-step approaches", the "border control" that prevents the normal bacterial flora on the ocular surface from entering the eye is the most important. As a "border control" method, we explain how to clean the eye surface every 20-30 seconds during cataract surgery with 0.25% povidone-iodine instead of washing with physiological saline.

The text in the original Introduction has been moved to a new section 2. Border control.

Line 55: Please provide a more complete definition of endophthalmitis. Endophthalmitis is the most serious and devastating condition within the eye globe, which may lead to irreversible blindness in the affected eye. It is a purulent inflammation of the intraocular fluids, i.e. the vitreous and the aqueous humor. Although the etiology of this condition can be both endogenous and exogenous, it is mostly secondary to intraocular surgeries, injections, or penetrating ocular trauma. Please also mention in a sentence the treatment of this serios complication to provide to readers a more complete overview of the topic.

Please use the following reference:

Urgent Vitrectomy with Vancomycin Infusion, Silicone Oil Endotamponade, and General Antibiotic Treatment in Multiple Cases of Endophthalmitis from a Single Day of Intravitreal Injections-Case Series. J Clin Med. 2021 Mar 4;10(5):1059. doi: 10.3390/jcm10051059. PMID: 33806541; PMCID: PMC7961493.

Response: We have added the following sentences as suggested by the reviewer and added one reference (lines 73-78):

“Endophthalmitis is the most serious and devastating condition within the eye globe, which may lead to irreversible blindness in the affected eye. It is a purulent inflammation of the intraocular fluids comprising the vitreous and the aqueous humor. Although the etiology of this condition can be both endogenous and exogenous, it is mostly secondary to intraocular surgeries, injections, or penetrating ocular trauma [9]. Postoperative endophthalmitis is treated ……”

  1. Pietras-Baczewska, A.; Jasińska, E.; Toro, M.D.; Bonfiglio, V.; Reibaldi, M.; Avitabile, T.; Nowomiejska, K.; Rejdak, R. Urgent vitrectomy with vancomycin infusion, silicone oil endotamponade, and general antibiotic treatment in multiple cases of endophthalmitis from a single day of intravitreal injections-case series. J. Clin. Med. 2021,10(5):1059.

Authors should underline more in detail that cataract surgery is the most prevalent surgery performed in ophthalmology. With a global prevalence of about 50% in adults over 50 years of age, cataract surgery is the most common operative procedure performed in developed countries.12 In 2010, of the total reported 32.4 million blind and 191 million vision-impaired cases, 10.8 million people were blind and 35.1 million were visually impaired due to cataract.

Please use the following reference:

Early impact of COVID-19 outbreak on eye care: Insights from EUROCOVCAT group. Eur J Ophthalmol. 2021 Jan;31(1):5-9. doi: 10.1177/1120672120960339. Epub 2020 Sep 24. PMID: 32967466.

Rethinking Elective Cataract Surgery Diagnostics, Assessments, and Tools after the COVID-19 Pandemic Experience and Beyond: Insights from the EUROCOVCAT Group. Diagnostics (Basel). 2020 Dec 2;10(12):1035. doi: 10.3390/diagnostics10121035. PMID: 33276612; PMCID: PMC7761628.

Response: We have added the following in Introduction (lines 27-32):

“Cataract surgery is the most common surgery performed in ophthalmology. With a global cataract prevalence of approximately 50% in adults over the age of 50, cataract surgery is the most common surgical procedure performed in developed countries. Of the overall 32.4 million visually impaired and 191 million visually impaired reported in 2010, 10.8 million people were blind and 35.1 million were visually impaired due to cataract [1].”

  1. Khairallah, M.; Kahloun, R.; Bourne, R.; Limburg, H.; Flaxman, S.R.; Jonas, J.B.; Keeffe, J.; Leasher, J.; Naidoo, K.; Pesudovs, K.; Price, H.; White, R.A.; Wong, T.Y.; Resnikoff, S.; Taylor, H.R. Vision Loss Expert Group of the Global Burden of Disease Study. Number of People Blind or Visually Impaired by Cataract Worldwide and in World Regions, 1990 to 2010. Invest. Ophthalmol. Vis. Sci. 2015,56(11):6762-6769.

We have added one suggested reference in Line 44.

“Coronavirus disease 2019 (COVID-19) is spreading all over the world [2].

2.Toro, M.D.; Brézin, A.P.; Burdon, M.; Cummings, A.B.; Evren, Kemer. O.; Malyugin, B.E.; Prieto, I.; Teus, M.A.; Tognetto, D.; Törnblom, R.; Posarelli, C.; ChorÄ…giewicz, T.; Rejdak, R. Early impact of COVID-19 outbreak on eye care: Insights from EUROCOVCAT group. Eur. J. Ophthalmol. 2021,31(1):5-9.

All ocular surgeries should be also considered as an ocular surface damaging event. Indeed, it is also known that the incidence and severity of dry eye symptoms may increase after cataract surgery. To achieve the best outcome in cataract surgery, a healthy ocular surface is crucial. Patients with more severe ocular surface disease are at higher risk of post-operative complications such as secondary infections.

Please use the following reference:

Prevalence of Ocular Demodicosis and Ocular Surface Conditions in Patients Selected for Cataract Surgery. J Clin Med. 2020 Sep 23;9(10):3069. doi: 10.3390/jcm9103069. PMID: 32977656; PMCID: PMC7598293.

Response: We have added the following sentences (lines 475-477):

“It is also known that the incidence and severity of dry eye symptoms may increase after cataract surgery. To achieve the best outcome in cataract surgery, a healthy ocular surface is crucial [71].”

  1. Nowomiejska, K.; Lukasik, P.; Brzozowska, A.; Toro, M.D.; Sedzikowska, A.; Bartosik, K.; Rejdak, R. Prevalence of ocular demodicosis and ocular surface conditions in patients selected for cataract surgery. J. Clin. Med. 2020,9(10):3069

However, povidone iodine antisepsis does not reduce to nil the risk of endophthalmitis after intravitreal therapy, since its incidence after PI application ranges from to 0.02% to 0.3% and a cumulative rate throughout the treatment series was reported in up to 1% of patients. Studies on conjunctival swab after PI antisepsis showed a significant reduction of bacterial load on the eye surface, but not a complete eradication, with a lowest rate of culture-positive swabs of 3%. Likewise, needles used for intravitreal injections, after povidone iodine antisepsis, have been found to be contaminated by bacteria, with a range varying from 0.4% to 21%.

Please discuss more in details these aspects using the following reference:

The Effectiveness of 0.6% Povidone Iodine Eye Drops in Reducing the Conjunctival Bacterial Load and Needle Contamination in Patients Undergoing Anti-VEGF Intravitreal Injection: A Prospective, Randomized Study. J Clin Med. 2019 Jul 13;8(7):1031. doi: 10.3390/jcm8071031. PMID: 31337003; PMCID: PMC6678890.

Response: In this review, we focus on cataract surgery. Therefore, we would like to omit detailed discussion of intravitreal injection. Regarding use of 0.25% povidone-iodine in intravitreal injections, our studies showed very low endophthalmitis incidence of 0/15,144 injections (Shimada et al. Minimizing the endophthalmitis rate following intravitreal injections using 0.25% povidone-iodine irrigation and surgical mask. Graefe's Arch Clin Exp Ophthalmol 2013; 251 (8): 1185-1189) and 0/12,523 injections (Tanaka K, et al. No increase in incidence of post-intravitreal injection endophthalmitis without topical antibiotics : a prospective study. Jpn J Ophthalmol. 2019; 63 (5): 396-401). This is probably because we wash with 0.25% povidone-iodine before, immediately before, and immediately after the injection.

Line 67-68: Therefore, it is necessary to prevent endophthalmitis using drugs or other agents with a wide antimicrobial spectrum against also multidrug-resistant bacteria and fungi.

Paduch R, Urbanik-Sypniewska T, Kutkowska J, ChorÄ…giewicz T, Matysik-Woźniak A, Zweifel S, Czarnek-Chudzik A, ZaÅ‚uska W, Rejdak R, Toro MD. Ozone-Based Eye Drops Activity on Ocular Epithelial Cells and Potential Pathogens Infecting the Front of the Eye. Antioxidants. 2021; 10(6):968. https://doi.org/10.3390/antiox10060968.         

Response: As suggested, we have included this reference (line 238-241):

“Ozonated solution (4 ppm) is effective for disinfecting the eye surface [39, 40], but has several issues including short half-life, need for ozone generator, and damage to corneal endothelial cells after exposure for over 10 seconds [41].”

  1. Paduch, R.; Urbanik-Sypniewska, T.; Kutkowska, J.; ChorÄ…giewicz, T.; Matysik-Woźniak, A.; Zweifel, S.; Czarnek-Chudzik, A.; ZaÅ‚uska, W.; Rejdak, R.; Toro, M.D. Ozone-Based Eye Drops Activity on Ocular Epithelial Cells and Potential Pathogens Infecting the Front of the Eye. Antioxidants. 2021,10(6):968.                          

“Continuous use of the same antibiotic eye drops induces growth of bacterial species re-92 sistant to the eye drops by the microbial substitution phenomenon. When administration 93 of postoperative antibiotic eye drops is finished, instead of recovery of the bacterial flora 94 to the preoperative state, there is an increase of resistant bacteria. 95”. Please add a reference.7

Response: We have cited some references as follows (line 116-118):

“Continuous use of the same antibiotic eye drops induces growth of bacterial species resistant to the eye drops by the microbial substitution phenomenon [22-24].”

  1. Matsuura, K.; Miyazaki, D.; Sasaki, S.I.; Inoue, Y.; Sasaki, Y.; Shimizu, Y. Effectiveness of intraoperative iodine in cataract surgery: cleanliness of the surgical field without preoperative topical antibiotics. Jpn. J. Ophthalmol. 2020,64(1),37-44.
  2. Hsu, J.; Gerstenblith, A.T.; Garg, S.J.; Vander, J.F. Conjunctival flora antibiotic resistance patterns after serial intravitreal injections without postinjection topical antibiotics. Am. J. Ophthalmol. 2014,157(3),514-518.e1
  3. Storey, P.; Dollin, M.; Rayess, N.; Pitcher, J.; Reddy, S.; Vander, J.; Hsu, J.; Garg, S. Post-Injection Endophthalmitis Study Team. The effect of prophylactic topical antibiotics on bacterial resistance patterns in endophthalmitis following intravitreal injection. Graefes. Arch. Clin. Exp. Ophthalmol. 2016,254(2),235-242.

All Figures need a title and a more clear explanation of what is shown in the figures.

Response: In the legend of Figure 10, we have added “instead of washing with saline”, and “The applied 0.25% povidone-iodine flows from the cornea to the conjunctiva, so the visibility of the anterior chamber is good”. The legends of the other figures are already lengthy. We have not added more explanations.

Line 103-105: “Since there 103 is still a risk that a minute number of bacteria may remain in the anterior chamber at the 104 end of surgery, postoperative antibiotic eye drops are considered necessary.” Are you sure? Please add a reference.

Response: We have changed the sentence to the following (lines 130-133):

“Postoperative antibiotic eye drops are expected to penetrate the anterior chamber and exhibit antibacterial effects. In many countries and facilities, postoperative antibiotic eye drops are used from less than 7 days to more than 30 days (median 7-13 days) [18].

  1. Grzybowski, A.; Schwartz, S.G.; Matsuura, K.; et al. Endophthalmitis Prophylaxis in Cataract Surgery: Overview of current practice patterns around the world. Curr. Pharm. Des. 2017;23(4):565-573.

The paper needs a professional English Editing and the entire flow needs to be improved.

Response: The revised manuscript has been edited by a professional editor.

Round 2

Reviewer 1 Report

The authors answered to all my comments, but I have serious concerns, as the Conclusio still claims that the proposed method “effectively reduces the risk of postoperative endophthalmitis”. There is just no evidence for this conclusion. Furthermore, I think that there is still too little attention paid to the discussion of current state of the art in endophthalmitis prophylaxis.

Line 31: remove “.” and replace P with p

Line 36: I think this sentence should be changed, as there is – as written in the last report – no evidence, if the approach prevents postoperative endophthalmitis.

Line 44-51: I recommend sparing the comparison to the Covid pandemic out.

Line 76: “Postoperative endophthalmitis is treated.” – This sentence is probably uncompleted.

Line 76-78: There is still no correlation between incidence and prognosis.

Line 103: I think the term “prophylaxis” suits better than “disinfection”.

Line 448: If the 0.25% povidone-iodine flows from the cornea to the conjunctiva for good visibility, how can application time be guaranteed? I think this needs to be discussed.

Conclusion: “Using 5-10% povidone-iodine for skin disinfection before surgery and repetitive washing of the ocular surface with 0.25% povidone-iodine every 30 seconds throughout cataract surgery effectively reduce the risk of postoperative endophthalmitis.” – As stated in the last report, without any RCT, it must not be stated that this method effectively reduces the risk of postoperative endophthalmitis.

“We have cited the following two papers.

  1. Endophthalmitis Study Group, European Society of Cataract & Refractive Surgeons. Prophylaxis of postoperative endophthalmitis following cataract surgery: results of the ESCRS multicenter study and identification of risk factors. J. Cataract. Refract. Surg.

2007,33(6),978-988.

  1. Bowen, R.C.; Zhou, A.X.; Bondalapati. S.; Lawyer, T.W.; Snow, K.B.; Evans, P.R.; Bardsley, T.; McFarland, M.; Kliethermes, M.; Shi, D.; Mamalis, C.A.; Greene, T.; Rudnisky, C.J.; Ambati, B.K. Comparative analysis of the safety and efficacy of intracameral cefuroxime, moxifloxacin and vancomycin at the end of cataract surgery: a meta-analysis. Br. J. Ophthalmol2018,102(9),1268-1276.“

It is not about just citing the papers, they need to be discussed. What does the method add to effect of intracameral antibiosis? What is the risk-benefit-assessment in comparison?

Author Response

Comments and Suggestions for Authors

The authors answered to all my comments, but I have serious concerns, as the Conclusion still claims that the proposed method “effectively reduces the risk of postoperative endophthalmitis”. There is just no evidence for this conclusion. Furthermore, I think that there is still too little attention paid to the discussion of current state of the art in endophthalmitis prophylaxis.

Response: We understand the reviewer’s concern about the conclusion. We apologize for the overstated claim. We have changed the expression as follows:

Line 575: “effectively reduces the risk of postoperative endophthalmitis” changed to” expected to be a preventive method for endophthalmitis”.

Line 31: remove “.” and replace P with p

Response: We apologize for the typographical error. We have made the following change:

Line 31: “….. reported in 2010, 10.8 million people were blind and 351 million were …….”

Line 36: I think this sentence should be changed, as there is – as written in the last report – no evidence, if the approach prevents postoperative endophthalmitis.

Response: Again we apologize for the overstatement. We have made the following change:

Line 37-38: “In this paper, we introduce a "3-step approach" to prevent post-cataract endophthalmitis caused by intraocular migration of bacteria during cataract surgery.” changed to “This article introduces the concept of a "three-step approach" to prevent endophthalmitis.”

Line 44-51: I recommend sparing the comparison to the Covid pandemic out.

Response: Following the reviewer’s recommendation, we have omitted COVID-19 and replace with microorganisms in general, as follows:

Lines 44-46: “Coronavirus disease 2019 (COVID-19) is spreading all over the world [2]. Once the coronavirus succeeds to gain foothold in a country, it is difficult to control and eliminate it, which makes us keenly aware of the importance of “border control” changed to “In a pandemic, once the causative bacteria or viruses succeed to gain foothold in a country, it is difficult to control and eliminate them [6-9], which makes us keenly aware of the importance of “border control”.

We have cited the four references that were recommended by the other reviewer.

Line 76: “Postoperative endophthalmitis is treated.” – This sentence is probably uncompleted.

Response: We apologize for the incomplete sentence.

Lines 75-76: ”Postoperative endophthalmitis is treated.” has been deleted.

Line 76-78: There is still no correlation between incidence and prognosis.

Response: We agree that there is no correlation between incidence and prognosis. We have made the following change:

Lines 77-78: “Although the incidence of endophthalmitis after cataract surgery is low with reports of 0.076% [10], 0.02% [11] and 0.048% [12], the prognosis is unfavorable.” changed to “The incidence of endophthalmitis after cataract surgery is low with reports of 0.076% [10], 0.02% [11] and 0.048% [12].”

Line 103: I think the term “prophylaxis” suits better than “disinfection”.

Response: We agree. We have changed the sentence as follows:

Line 99: “Many different practice patterns of infection prophylaxis have been reported around the world [17, 18].” changed to “Many different practice patterns of disinfection have been reported around the world [17, 18].”

Line 448: If the 0.25% povidone-iodine flows from the cornea to the conjunctiva for good visibility, how can application time be guaranteed? I think this needs to be discussed.

Response: As stated in the preceding sentence, the cornea is irrigated repeatedly every 20-30 seconds. Fresh 0.25% povidone-iodine is applied to the cornea before the corneal surface becomes dry due to flow of 0.25% povidone-iodine to the conjunctiva. We have added the following sentences.

Lines 441-445: “Since 0.25% povidone-iodine shows a bactericidal effect within 15 seconds,” “Fresh 0.25% povidone-iodine is applied to the cornea before the corneal surface becomes dry due to flow of 0.25% povidone-iodine to the conjunctiva.”

Lines 450-451: “In the povidone-iodine group, the ocular surface was washed with 0.25% povidone-iodine every 20-30 seconds.”

Conclusion: “Using 5-10% povidone-iodine for skin disinfection before surgery and repetitive washing of the ocular surface with 0.25% povidone-iodine every 30 seconds throughout cataract surgery effectively reduce the risk of postoperative endophthalmitis.” – As stated in the last report, without any RCT, it must not be stated that this method effectively reduces the risk of postoperative endophthalmitis.

Response: We understand the reviewer’s concern. We apologize for the overstated claim. We have changed the conclusion as follows:

Line 571-575: “Povidone-iodine has several advantages including established effective and safe concentrations for ocular tissues, broad-spectrum antimicrobial activity, short exposure time for microbial killing, absence of resistant strains, low cost, and easy to use worldwide. Using 5-10% povidone-iodine for skin disinfection before surgery and repetitive washing of the ocular surface with 0.25% povidone-iodine every 30 seconds throughout cataract surgery effectively reduce the risk of postoperative endophthalmitis.” changed to

“Povidone-iodine has several advantages including established effective and safe concentrations for ocular tissues, broad-spectrum antimicrobial activity, short exposure time for microbial killing, absence of resistant strains, low cost, and easy to use worldwide. Repetitive washing of the ocular surface with 0.25% povidone-iodine every 20-30 seconds throughout cataract surgery is expected to be a preventive method for postoperative endophthalmitis.”

We have cited the following two papers.

  1. Endophthalmitis Study Group, European Society of Cataract & Refractive Surgeons. Prophylaxis of postoperative endophthalmitis following cataract surgery: results of the ESCRS multicenter study and identification of risk factors. J. Cataract. Refract. Surg.

2007,33(6),978-988.

  1. Bowen, R.C.; Zhou, A.X.; Bondalapati. S.; Lawyer, T.W.; Snow, K.B.; Evans, P.R.; Bardsley, T.; McFarland, M.; Kliethermes, M.; Shi, D.; Mamalis, C.A.; Greene, T.; Rudnisky, C.J.; Ambati, B.K. Comparative analysis of the safety and efficacy of intracameral cefuroxime, moxifloxacin and vancomycin at the end of cataract surgery: a meta-analysis. Br. J. Ophthalmol. 2018,102(9),1268-1276.“

It is not about just citing the papers, they need to be discussed. What does the method add to effect of intracameral antibiosis? What is the risk-benefit-assessment in comparison?

Additional revision

We have added a new section on intracameral antibiotics as follows:

Lines 550-560:

11. Intracameral antibiotics as prophylaxis of postoperative endophthalmitis

    A prospective randomized partially masked multicenter cataract surgery study in nine European countries which recruited 16,603 patients concluded that use of intracameral cefuroxime at the standard dose of 1 mg/0.1 ml at the end of surgery reduced the incidence of postoperative endophthalmitis [37]. Furthermore, a meta-analysis reported that intracameral cefuroxime and moxifloxacin at standard doses reduced endophthalmitis rates compared to controls with minimal or no toxicity events [89]. However, use of vancomycin has been reported to be associated with hemorrhagic occlusive retinal vasculitis [90], and postoperative endophthalmitis due to cefuroxime-resistant strains despite intracameral antibiotic prophylaxis is an issue [91,92]. Combined use of standard concentrations of intracameral antibiotics and 0.25% povidone-iodine ocular surface irrigation may be a strategy to be explored.

References

  1. Endophthalmitis Study Group, European Society of Cataract & Refractive Surgeons. Prophylaxis of postoperative endophthalmitis following cataract surgery: results of the ESCRS multicenter study and identification of risk factors. J. Cataract. Refract. Surg. 2007,33(6),978-988.
  2. Bowen, R.C.; Zhou, A.X.; Bondalapati, S.; Lawyer, T.W.; Snow, K.B.; Evans, P.R.; Bardsley, T.; McFarland, M.; Kliethermes, M.; Shi, D.; Mamalis, C.A.; Greene, T.; Rudnisky, C.J.; Ambati, B.K. Comparative analysis of the safety and efficacy of intracameral cefuroxime, moxifloxacin and vancomycin at the end of cataract surgery: a meta-analysis. Br. J. Ophthalmol, 2018,102(9),1268-1276.
  3. Witkin, A.J.; Shah, A.R.; Engstrom, R.E.; Kron-Gray, M.M.; Baumal, C.R.; Johnson, M.W.; Witkin, D.I.; Leung, J.; Albini, T.A.; Moshfeghi, A.A.; Batlle, I.R.; Sobrin, L.; Eliott, D. Postoperative hemorrhagic occlusive retinal vasculitis: expanding the clinical spectrum and possible association with vancomycin. Ophthalmology. 2015,122(7):1438-1451.92. Mesnard, C.; Beral, L.; Hage, R.; Merle, H.; Farès, S.; David, T.
  4. Endophthalmitis after cataract surgery despite intracameral antibiotic prophylaxis with licensed cefuroxime. J. Cataract. Refract. Surg. 2016,42(9):1318-1323.
  5. Shorstein, N.H.; Liu, L.; Carolan, J.A.; Herrinton, L. Endophthalmitis prophylaxis failures in patients injected with intracameral antibiotic during cataract surgery. Am. J. Ophthalmol. 2021,227(7):166-172. 

Reviewer 2 Report

The paper has been improved during the review process and authors improved the quality of the flow. Additionally they replaied to almost all my previous queries. However, I still have some major concerns related to the content of the manuscript that need to be up-dated and clarify before the paper is suitable for pubblication.

I hope the authors will find usefull my suggestions reported below. Looking to receive your paper once again after their revision.

1) Still many paragraph of the papers have no a reference. Each sentence or statement need to have a reference. For example in the introduction… “Among the complications of cataract surgery, postoperative endophthalmitis is the most serious condition, and prevention of postoperative endophthalmitis is an important issue. Postoperative endophthalmitis after cataract surgery is typically caused by the patient's own conjunctival normal bacterial flora. The current problem is an increase of multidrug-resistant bacteria and fungi causing endophthalmitis.”

2) The reference N° 2 related to EUROCOVCAT group needs to be placed at the end of fisrt and second sentence of the introduction as well.

3) The authors state that Coronavirus disease 2019 (COVID-19) is spreading all over the world. I would suggest to add that it is still impacting all the field of ophthalmology and eye care in general with drammatic consequences for the health care Systems. To clarify these aspects discussed in the paper listed as reference number 2 will help readers to understand better why is important to continue to treat our patients even in case of a pandemic.

4) From line 44 to line 50 of the paragraph related to the border control tha authors state that “Once the coronavirus succeeds to gain foothold in a country, it is difficult to control and eliminate it, which makes us keenly aware of the importance of “border control”. Closing the borders, strengthening quarantine measures at airports and ports, and blocking the flow of people infected with the virus from entering one’s country are very useful measures. Regarding the threat and control of infection, both countries surrounded by sea and countries bordering other countries are in a situation not unlike that of an eyeball surrounded by surface fluid.”

Please clarify that while some restrictions and new issues were taken to arrest the spread of the COVID-19 infection inside the society, in the field of ophthalmology many scientific societies and comittee proposed new guidelines and suggestions how to rethink elective cataract surgery diagnostics, assessments, and tools after the covid-19 pandemic experience, with a special focus on the control of the infection during surgery.  Otherwise, the entire paragraph in meaningless.

Please add these aspect and cite the reference reported below:

  • Tognetto et al. Rethinking Elective Cataract Surgery Diagnostics, Assessments, and Tools after the COVID-19 Pandemic Experience and Beyond: Insights from the EUROCOVCAT Group. Diagnostics (Basel). 2020 Dec 2;10(12):1035.

5) Please also clarify that coronavirus SARS-CoV-2 responsible for the current human COVID-19 pandemic has shown tropism toward different organs with variable efficiency, eyes included. To add these aspects will help the justify the theory of the border control and its role in the prevention in general.

Please use the following two references:

  • Toro M, et al. Early Impact of COVID-19 Outbreak on the Availability of Cornea Donors: Warnings and Recommendations. Clin Ophthalmol. 2020 Sep 25;14:2879-2882.
  • Dolar-Szczasny J, et al. Ocular Involvement of SARS-CoV-2 in a Polish Cohort of COVID-19-Positive Patients. Int J Environ Res Public Health. 2021 Mar 12;18(6):2916.

6) Some authors have reported that povidone iodine (PI) antisepsis does not reduce the risk of endophthalmitis after intravitreal therapy, since its incidence after PI application ranges from to 0.02% to 0.3% and a cumulative rate throughout the treatment series was reported in up to 1% of patients. Studies on conjunctival swab after PI antisepsis showed a significant reduction of bacterial load on the eye surface, but not a complete eradication, with a lowest rate of culture-positive swabs of 3%. Likewise, needles used for intravitreal injections, after povidone iodine antisepsis, have been found to be contaminated by bacteria, with a range varying from 0.4% to 21%.

These aspect need to be adressed. PI has show a safer and a higher effectiveness regarding its antiseptic profile in the prevention of endophthalmitis following ocular surgery.

Please discuss more in details these aspects using the following reference:

  • The Effectiveness of 0.6% Povidone Iodine Eye Drops in Reducing the Conjunctival Bacterial Load and Needle Contamination in Patients Undergoing Anti-VEGF Intravitreal Injection: A Prospective, Randomized Study. J Clin Med. 2019 Jul 13;8(7):1031.

Author Response

Comments and Suggestions for Authors

The paper has been improved during the review process and authors improved the quality of the flow. Additionally they replaied to almost all my previous queries. However, I still have some major concerns related to the content of the manuscript that need to be up-dated and clarify before the paper is suitable for pubblication.

I hope the authors will find usefull my suggestions reported below. Looking to receive your paper once again after their revision.

  • Still many paragraph of the papers have no a reference. Each sentence or statement need to have a reference. For example in the introduction… “Among the complications of cataract surgery, postoperative endophthalmitis is the most serious condition, and prevention of postoperative endophthalmitis is an important issue. Postoperative endophthalmitis after cataract surgery is typically caused by the patient's own conjunctival normal bacterial flora. The current problem is an increase of multidrug-resistant bacteria and fungi causing endophthalmitis.”

Response: We thank the reviewer for helping us improve our manuscript. We have added references to the paragraph as follows:

Lines 32-36: Changed to ”Among the complications of cataract surgery, postoperative endophthalmitis is one of the most serious condition [2], and prevention of postoperative endophthalmitis is an important issue. Postoperative endophthalmitis after cataract surgery is typically caused by the patient's own conjunctival normal bacterial flora [3]. The current problem is an increase of multidrug-resistant bacteria and fungi causing endophthalmitis [4, 5].”

References

  1. Kresloff, M.S.; Castellarin, A.A.; Zarbin, M.A. Surv. Ophthalmol, 1998,43(3),193-224.
  2. Speaker, M.G.; Milch, F.A.; Shah, M.K.; Eisner, W.; Kreiswirth, B.N. Role of external bacterial flora in the pathogenesis of acute postoperative endophthalmitis. Ophthalmology. 1991,98(5),639-649.
  3. Yannuzzi, N.A.; Si, N.; Relhan, N.; Kuriyan, A.E.; Albini, T.A.; Berrocal, A.M.; Davis JL, Smiddy WE, Townsend J, Miller D, Flynn HW Jr. Endophthalmitis after clear corneal cataract surgery: Outcomes over two decades. J. Ophthalmol, 2017, 174,155-159.
  4. Smith, T.C.; Benefield. R,J.; Kim, J.H. Risk of fungal endophthalmitis associated with cataract surgery: A mini-review. 2015,180(5-6),291-297.

  • The reference N° 2 related to EUROCOVCAT group needs to be placed at the end of fisrt and second sentence of the introduction as well.
  • The authors state that Coronavirus disease 2019 (COVID-19) is spreading all over the world. I would suggest to add that it is still impacting all the field of ophthalmology and eye care in general with drammatic consequences for the health care Systems. To clarify these aspects discussed in the paper listed as reference number 2 will help readers to understand better why is important to continue to treat our patients even in case of a pandemic.
  • From line 44 to line 50 of the paragraph related to the border control tha authors state that “Once the coronavirus succeeds to gain foothold in a country, it is difficult to control and eliminate it, which makes us keenly aware of the importance of “border control”. Closing the borders, strengthening quarantine measures at airports and ports, and blocking the flow of people infected with the virus from entering one’s country are very useful measures. Regarding the threat and control of infection, both countries surrounded by sea and countries bordering other countries are in a situation not unlike that of an eyeball surrounded by surface fluid.”

Please clarify that while some restrictions and new issues were taken to arrest the spread of the COVID-19 infection inside the society, in the field of ophthalmology many scientific societies and comittee proposed new guidelines and suggestions how to rethink elective cataract surgery diagnostics, assessments, and tools after the covid-19 pandemic experience, with a special focus on the control of the infection during surgery.  Otherwise, the entire paragraph in meaningless.

Please add these aspect and cite the reference reported below:

  • Tognetto et al. Rethinking Elective Cataract Surgery Diagnostics, Assessments, and Tools after the COVID-19 Pandemic Experience and Beyond: Insights from the EUROCOVCAT Group. Diagnostics (Basel). 2020 Dec 2;10(12):1035.
  • Please also clarify thatcoronavirus SARS-CoV-2 responsible for the current human COVID-19 pandemic has shown tropism toward different organs with variable efficiency, eyes included. To add these aspects will help the justify the theory of the border control and its role in the prevention in general.

Please use the following two references:

  • Toro M, et al. Early Impact of COVID-19 Outbreak on the Availability of Cornea Donors: Warnings and Recommendations. Clin Ophthalmol. 2020 Sep 25;14:2879-2882.
  • Dolar-Szczasny J, et al. Ocular Involvement of SARS-CoV-2 in a Polish Cohort of COVID-19-Positive Patients. Int J Environ Res Public Health. 2021 Mar 12;18(6):2916.Dolar-Szczasny J, et al. Ocular Involvement of SARS-CoV-2 in a Polish Cohort of COVID-19-Positive Patients. Int J Environ Res Public Health. 2021 Mar 12;18(6):2916.Toro M, et al. Early Impact of COVID-19 Outbreak on the Availability of Cornea Donors: Warnings and Recommendations. Clin Ophthalmol. 2020 Sep 25;14:2879-2882.

Response: Comments 2) to 5) concern COVID-19 with respect to border control. We would like to respond to these comments together. We greatly appreciate the reviewer for the insightful comments and advice on this topic, which would have added depth to this review. However, we have a problem with addressing these comments.

The reason is that another reviewer has been suggesting that we omit the discussion of COVID-19. In the first review, we received a comment: "In my personal opinion, the comparison to CoVid-19 pandemic as an opening statement is misleading in the context of the manuscript". To address this comment, we restructured Introduction and mentioned COVID-19 in the second section only once. However, in the second review, we received another comment: “I recommend sparing the comparison to the covid pandemic out”. Since readers may mistakenly think that COVID-19 causes endophthalmitis, we decided to omit “COVID-19 disease” and replace with microorganisms in general, as follows:

Lines 44-46: “Coronavirus disease 2019 (COVID-19) is spreading all over the world [2]. Once the coronavirus succeeds to gain foothold in a country, it is difficult to control and eliminate it, which makes us keenly aware of the importance of “border control” changed to “In a pandemic, once the causative bacteria or viruses succeed to gain foothold in a country, it is difficult to control and eliminate them [6-9], which makes us keenly aware of the importance of “border control”.

We have cited the three new references recommended by this reviewer.

  1. Tognetto, D.; Brézin, A.P.; Cummings, A.B.; Malyugin, B.E.; Evren, Kemer. O.; Prieto, I.; Rejdak, R.; Teus, M.A.; Törnblom, R.; Toro, M.D.; Vinciguerra, A.L.; Giglio, R.; De, Giacinto. C. Rethinking elective cataract surgery diagnostics, assessments, and tools after the COVID-19 pandemic experience and beyond: Insights from the EUROCOVCAT group. Diagnostics (Basel). 2020,10(12):1035
  2. Toro M, Choragiewicz T, Posarelli C, Figus M, Rejdak R; European COVID-19 Cataract Group (#EUROCOVCAT). Early impact of COVID-19 outbreak on the availability of cornea donors: Warnings and recommendations. Clin. Ophthalmol. 2020,14,2879-2882.
  3. Dolar-Szczasny, J.; Toro, M.D.; DworzaÅ„ska, A.; Wójtowicz, T.; Korona-Glowniak, I.; Sawicki, R.; Boguszewska, A.; Polz-Dacewicz, M.; Tomasiewicz, K.; ZaÅ‚uska, W.; Rejdak, R.; Bagnoli, P.; Rusciano, D. Ocular Involvement of SARS-CoV-2 in a Polish cohort of COVID-19-Positive patients. Int. J. Environ. Res. Public. Health. 2021,18(6),2916.

  • Some authors have reported that povidone iodine (PI) antisepsis does not reduce the risk of endophthalmitis after intravitreal therapy, since its incidence after PI application ranges from to 0.02% to 0.3% and a cumulative rate throughout the treatment series was reported in up to 1% of patients. Studies on conjunctival swab after PI antisepsis showed a significant reduction of bacterial load on the eye surface, but not a complete eradication, with a lowest rate of culture-positive swabs of 3%. Likewise, needles used for intravitreal injections, after povidone iodine antisepsis, have been found to be contaminated by bacteria, with a range varying from 0.4% to 21%.

These aspect need to be adressed. PI has show a safer and a higher effectiveness regarding its antiseptic profile in the prevention of endophthalmitis following ocular surgery.

Please discuss more in details these aspects using the following reference:

  • The Effectiveness of 0.6% Povidone Iodine Eye Drops in Reducing the Conjunctival Bacterial Load and Needle Contamination in Patients Undergoing Anti-VEGF Intravitreal Injection: A Prospective, Randomized Study. J Clin Med. 2019 Jul 13;8(7):1031.

Response: To address this point, we have added the following text.

Lines 474-490:

8.4 Effectiveness on intravitreal injection

In intravitreal injection, bacteria are introduced into the vitreous via the needle tip [77]. A study evaluated the effectiveness of preservative-free 0.6% povidone iodine eye drops as perioperative prophylactic treatment in patients undergoing intravitreal injection randomized to a group receiving 0.6% povidone iodine eye drops for three day before injection, and a control group receiving placebo. Bacterial growth from conjunctival swab cultures was significantly lower after 0.6% povidone iodine prophylaxis compared to baseline and to placebo prophylaxis (p < 0.001). However, the bacteria eradication rate in the 0.6% povidone iodine group was 82%, and did not achieve 100% [78].

The effect of 0.6% povidone iodine eye drops shows a rapidly bactericidal effect [79, 80], Considering that bacteria are present in the folds of the conjunctiva, povidone iodine irrigation of the ocular surface is more effective than instillation on the conjunctiva [81]. For intravitreal injections, it is better to wash with a few ml of 0.6% povidone iodine shortly before, just before, and immediately after the injection. Using 0.25% povidone-iodine to irrigate the conjunctival sac both before and after injection, we reported 0 case of suspected or proven infectious endophthalmitis in 12,523 intravitreal injections (95% confidence interval 0 to 0.00024%) [82].

References

  1. Nakashizuka, H.; Shoji, J.; Shimada, H.; Yuzawa, M. Experimental visualization and quantification of vitreous contamination following intravitreal injections, Retina. 2016,36(10):1882-1887.
  2. Reibaldi, M.; Avitabile, T.; Bandello, F.; Longo, A.; Bonfiglio, V.; Russo, A.; Castellino, N.; Rejdak, R.; Nowomiejska, K.; Toro, M.; Furino, C.; Cillino, S.; Fiore, T.; Cagini, C.; Grass,i P.; Musumeci, R.; Cocuzza, C.E.; Martinelli, M.; Fallico M. The effectiveness of 0.6% povidone iodine eye drops in reducing the conjunctival bacterial load and needle contamination in patients undergoing anti-VEGF Intravitreal Injection: A prospective, randomized study. J. Clin. Med. 2019,8(7),1031. 
  3. Musumeci, R.; Bandello, F.; Martinelli, M.; Calaresu, E.; Cocuzza, C.E. In vitro bactericidal activity of 0.6% povidone-iodine eye drops formulation. Eur. J. Ophthalmol. 2019,29(6):673-677.
  4. Pinna, A.; Donadu, M.G.; Usai, D.; Dore, S.; D'Amico-Ricci, G.; Boscia, F.; Zanetti, S. In vitro antimicrobial activity of a new ophthalmic solution containing povidone-iodine 0.6% (IODIM®). Acta. Ophthalmol, 2020,98(2):e178-e180
  5. Miño, de Kaspar. H.; Chang, R.T.; Singh, K.; Egbert, P.R.; Blumenkranz, M.S.; Ta, C.N. Prospective randomized comparison of 2 different methods of 5% povidone-iodine applications for anterior segment intraocular surgery. Arch. Ophthalmol, 2005,123(2):161-165. 
  6. Tanaka, K.; Shimada, H.; Mori, R.; Nakashizuka, H.; Hattori, T.; Okubo, Y. No increase in incidence of post-intravitreal injection endophthalmitis without topical antibiotics: a prospective study. Jpn. J. Ophthalmol. 2019,63(5),396-401.

Additional revision

We have added the following text:

Lines 520-526:

“A multicenter, nonrandomized, prospective, controlled study evaluated 0.66% povidone-iodine eye drops (IODIM®) as perioperative prophylactic treatment against the conjunctival bacterial flora of patients who underwent cataract surgery, and found that 0.66% povidone-iodine eye drops used for three days prior to cataract surgery effectively reduced the conjunctival bacterial load. The 0.66% povidone-iodine eye drops may represent a valid perioperative prophylactic antiseptic adjuvant treatment to protect the ocular surface from microbial contamination in preparation of the surgical procedure [87].”

Reference

  1. Musumeci, R.; Troiano, P.; Martinelli, M.; Piovella, M.; Carbonara, C.; Rossi, S.; Alessio, G.; Molteni, L.; Molteni, C.G.; Saderi, L.; Sotgiu, G.; Cocuzza, C.E. Effectiveness of 0.66% Povidone-Iodine Eye Drops on Ocular Surface Flora before Cataract Surgery: A Nationwide Microbiological Study. J. Clin. Med. 2021,10(10),2198.

Round 3

Reviewer 1 Report

As the authors replied to all my comments and revised the manuscript appropriately, I do not have any further comments.

Reviewer 2 Report

Authors replaied to all queries and addressed all my main concerns, Now the paper is suitable for pubblication.